# An *Aedes aegypti*-associated fungus increases susceptibility to dengue virus by modulating gut trypsin activity

**Yesseinia I Angleró-Rodríguez, Octavio AC Talyuli[†], Benjamin J Blumberg, Seokyoung Kang, Celia Demby, Alicia Shields, Jenny Carlson, Natapong Jupatanakul, George Dimopoulos***

W. Harry Feinstone Department of Molecular Microbiology and Immunology, Johns Hopkins Bloomberg School of Public Health, Baltimore, United States

**Abstract** Transmission of dengue virus (DENV) requires successful completion of the infection cycle in the *Aedes aegypti* vector, which is initiated in the midgut tissue after ingestion of an infectious blood meal. While certain *Ae. aegypti* midgut-associated bacteria influence virus infection, little is known about the midgut-associated fungi (mycobiota), and how its members might influence susceptibility to DENV infection. We show that a *Talaromyces* (*Tsp_PR*) fungus, isolated from field-caught *Ae. aegypti,* render the mosquito more permissive to DENV infection. This modulation is attributed to a profound down-regulation of digestive enzyme genes and trypsin activity, upon exposure to *Tsp_PR*-secreted factors. In conclusion, we show for the first time that a natural mosquito gut-associated fungus can alter *Ae. aegypti* physiology in a way that facilitates pathogen infection.

DOI: https://doi.org/10.7554/eLife.28844.001

*For correspondence:
gdimopo1@jhu.edu

Present address: [†]Instituto de Bioquímica Médica Leopoldo de Meis, Universidade Federal do Rio de Janeiro, Rio de Janeiro, Brazil

Competing interests: The authors declare that no competing interests exist.

## Introduction

Arthropod-borne viral diseases have an enormous global public health impact. While dengue virus (DENV) remains the most important arbovirus worldwide, the recent emergence of Chikungunya and Zika virus have exacerbated the impact of *Aedes*-transmitted diseases. The mosquito *Aedes aegypti* is the principal vector of DENV. Successful transmission of the virus requires completion of its infection cycle in the mosquito, beginning in the midgut tissue and eventually ending up in the saliva, through which it can be introduced into a new host.

The mosquito acquires the virus through a blood meal from an infected vertebrate host. The blood is digested in the insect's midgut through the enzymatic action of a variety of proteolytic enzymes and provides the necessary nutrients for egg production (*Noriega and Wells, 1999*). The proteolytic activity of the midgut impairs DENV infection, and specific trypsins are associated with this modulation of susceptibility (*Brackney et al., 2008*; *Molina-Cruz et al., 2005*). In the midgut lumen, the virus also encounters a variety of microorganism, which constitutes the vector midgut microbiota. Studies have shown that some bacterial species can inhibit DENV infection and other human pathogens (*Bahia et al., 2014*; *Dennison et al., 2014*; *Ramirez et al., 2012*, *2014*). However, the midgut microbiota also comprises fungi, but little is yet known about the mosquito's mycobiome (*Angleró-Rodríguez et al., 2016*; *da Costa and de Oliveira, 1998*; *da S Pereira et al., 2009*). Most studies have focused on entomopathogenic fungi for use in mosquito control (*Dong et al., 2012*; *Scholte et al., 2007*). Fungi are ubiquitous in the environment, especially in tropical regions, and studies have shown the association of filamentous fungi, including *Talaromyces* species with various vectors of human pathogens (*Akhoundi et al., 2012*; *da Costa and de Oliveira, 1998*; *da S Pereira et al., 2009*; *Jaber et al., 2016*; *Marti et al., 2007*).

In the present study, we have identified a *Talaromyces* species fungus from Puerto Rico (*Tsp_PR*) from the midgut of field-caught *Ae. aegypti* that renders the mosquitoes more susceptible to DENV infection, through transcriptional and enzymatic inhibition of trypsin enzymes.

## Results

### An *Ae. aegypti* gut-associated *Tsp_PR* fungus augments DENV infection of the mosquito midgut

Adult female *Aedes sp.* mosquitoes were collected in the dengue endemic Maunabo region of southeastern Puerto Rico. After surface sterilization of mosquitoes, the midguts were dissected and plated on agar for fungal growth. Among the isolated fungi, we identified a *Talaromyces* species fungus and characterized at the microscopic (*Figure 1*) and molecular levels. Sequence analysis using the rRNA internal transcribed spacer (ITS) showed a 100% similarity to the fungus *Talaromyces* species and 99% *Penicillium* species. *Talaromyces* is often classified as a *Penicillium* species classification, since in 2011, species in the *Penicillium* subgenus *Biverticillium* were reclassified as *Talaromyces* (*Pitt, 2014*). To address the ability of *Tsp_PR* to colonize the mosquito midgut, we fed mosquitoes on fungus spore-laced sugar solution for 2 days, and then monitored the total fungus colony-forming units (CFUs) in the whole mosquito and midguts for 25 days (*Figure 2A,B*, *Figure 2—source data 1*). *Tsp_PR* was detected in the whole fungus-exposed mosquitoes and midguts during the entire time course. The assay also identified a small number of fungi in the control mosquito cohort but not *Tsp_PR*. We also investigated whether the presence of the fungus in the mosquito gut had any effect on mosquito longevity. Introduction of the live *Tsp_PR* spores through sugar feeding for 2 days did not affect mosquito survival up to 38 days (p=0.3073) (*Figure 2C*, *Figure 2—source data 1*).

To investigate whether the presence of *Tsp_PR* in the mosquito midgut can modulate susceptibility to DENV infection, we introduced whole fungus spores to *Ae. aegypti* mosquitoes via feeding on a spore-containing sugar solution prior to a DENV-infected blood meal (*Figure 2C,D*, *Figure 2—source data 1*). Two mosquito strains were used: the DENV-susceptible Rockefeller strain and the partly resistant Orlando strain (*Sim et al., 2013*). Seven days after DENV infection, midguts were dissected, and DENV titers were enumerated. Spore ingestion by Rockefeller and Orlando strains females resulted in a significant (p<0.001) increased DENV infection. Upon spore ingestion, both mosquito strains showed an increased DENV infection prevalence, as a measure of the proportion of mosquitoes that were infected, however it was only significantly higher in the Orlando strain. Further studies were performed in the Orlando strain.

*Talaromyces sp.* and related fungi are known to produce a variety of secondary metabolites and proteins (*Bara et al., 2013*; *Klitgaard et al., 2014*). It was unclear whether the influence of *Tsp_PR* on *Ae. aegypti's* susceptibility to DENV required the presence of whole live fungi or was mediated by a secreted factor. To distinguish between these two possibilities, we prepared a sucrose suspension of the *Tsp_PR* secretome containing a filtered fungus culture without spores and mycelia. Mosquitoes were fed on this suspension prior to the ingestion of DENV-infected blood (*Figure 3A*, *Figure 3—source data 1*). In a separate experiment, mosquitoes were fed on a sugar solution containing a heat-treated fungus secretome preparation prior to ingestion of DENV-infected blood, in order to address the nature of the factor(s) that influence DENV infection (*Figure 3B*, *Figure 3—source data 1*). Mosquitoes that had ingested the fungus secretome-containing solution showed a significantly increased virus infection (p=0.002). In contrast, the heat-treated solution did not enhance DENV infection of the mosquito midgut (p=0.548). Taken together, these findings indicate that *Tsp_PR* secretes one or more heat-sensitive molecules that enhance DENV infection in *Ae. aegypti* midguts.

Mosquito are exposed to a variety of fungi in nature. In order to test whether the *Tsp_PR*-mediated increase of DENV infection is fungus-specific, we performed fungus-exposure and DENV infection assays with a *Penicillium chrysogenum* that had also been isolated from field-caught mosquitoes. Feeding mosquitoes on whole *P. chrysogenum* conidia, or culture filtrates, did not affect DENV infection (p=0.829, p=0.867, respectively) (*Figure 4A,B*, *Figure 4—source data 1*). We have previously associated *P. chrysogenum* with an enhancement of *Plasmodium* infection in *Anopheles gambiae* (*Angleró-Rodríguez et al., 2016*). To test whether *Tsp_PR* could influence malaria parasite infection in its vector we provided *Tsp_PR* to *An. gambiae* mosquitoes prior to infection with

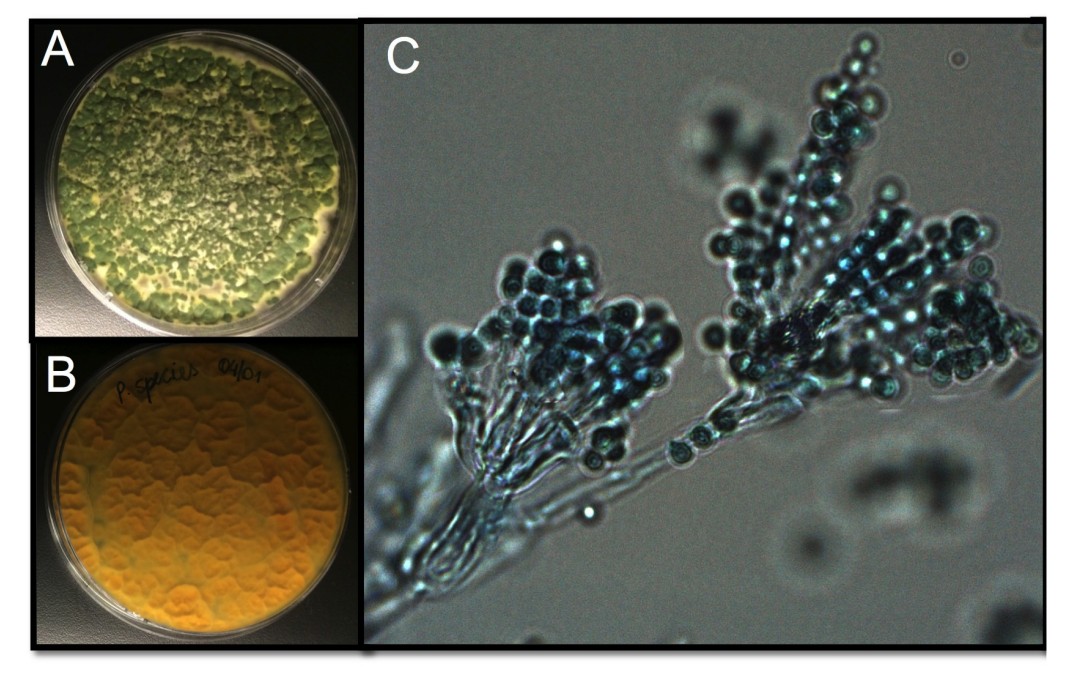

**Figure 1.** *Tsp_PR* morphology. After isolation, the fungus was grown on Sabouraud agar and characterized macroscopically and microscopically. (**A**) Top and (**B**) bottom view of the fungus on Sabouraud agar. (**C**) Microscopic view of the typical brush-like biverticillated conidiophore of *Talaromyces sp.* fungi.

DOI: https://doi.org/10.7554/eLife.28844.002

*Plasmodium falciparum,* and this resulted in increases parasite oocysts numbers on the mosquito gut (p=0.015) (*Figure 4C*, *Figure 4—source data 1*). These results show that *Tsp_PR* can enhance infection of different human pathogens in different mosquito vectors.

### *Tsp_PR*-secreted factor(s) do not affect the mosquito midgut microbiota or DENV infection of aseptic mosquitoes

Some species of *Tsp_PR* and the closely related *Penicillium* are known to produce bioactive compounds with anti-bacterial activity (*Bara et al., 2013*; *Klitgaard et al., 2014*). To investigate whether *Tsp_PR* produces antibacterial factors, we performed a bacterial growth inhibition assay by a disc diffusion antibiotic sensitivity test (*Figure 5A*). We examined possible *Tsp_PR*-mediated growth inhibition of the following bacteria isolated from field-caught mosquito midguts (*Ramirez et al., 2012*): the three Gram-negative bacteria *Serratia marcescens*, *Chromobacterium haemolyticum*, and *Enterobacter hormaechei*; and the three Gram-positive bacteria *Bacillus subtilis*, *Staphylococcus capprae*, and *Lactococcus lactis*. The disk soaked in the *Tsp_PR* secretome solution did not show any growth inhibitory activity for any of the six tested bacteria, when compared to an antibiotic control.

To investigate whether *Tsp_PR*-secreted molecules can influence the mosquito midgut bacterial load, we quantified the total cultivable bacteria of the midguts of fungus secretome-exposed and non-exposed mosquitoes using CFU assays (*Figure 5B*, *Figure 5—source data 1*). Exposure to the *Tsp_PR* secretome did not affect the total bacterial midgut load (p=0.147). Previous studies showed that reduction or elimination of the mosquito midgut microbiota, through antibiotic-treatment, (aseptic mosquitoes) significantly increases susceptibility to DENV in *Ae. aegypti* (*Xi et al., 2008*). Next, we tested whether the fungus-secretome would influence DENV infection of aseptic mosquitoes (*Figure 5C*, *Figure 5—source data 1*). *Tsp_PR* secretome-fed aseptic mosquitoes did not differ from the aseptic control cohort with regards to DENV infection intensity (p=0.867), while it showed a slightly higher but non-significant, infection prevalence. These data suggest that a bacteria-related factor may in some way influence the fungus-mediated effect on DENV infection.

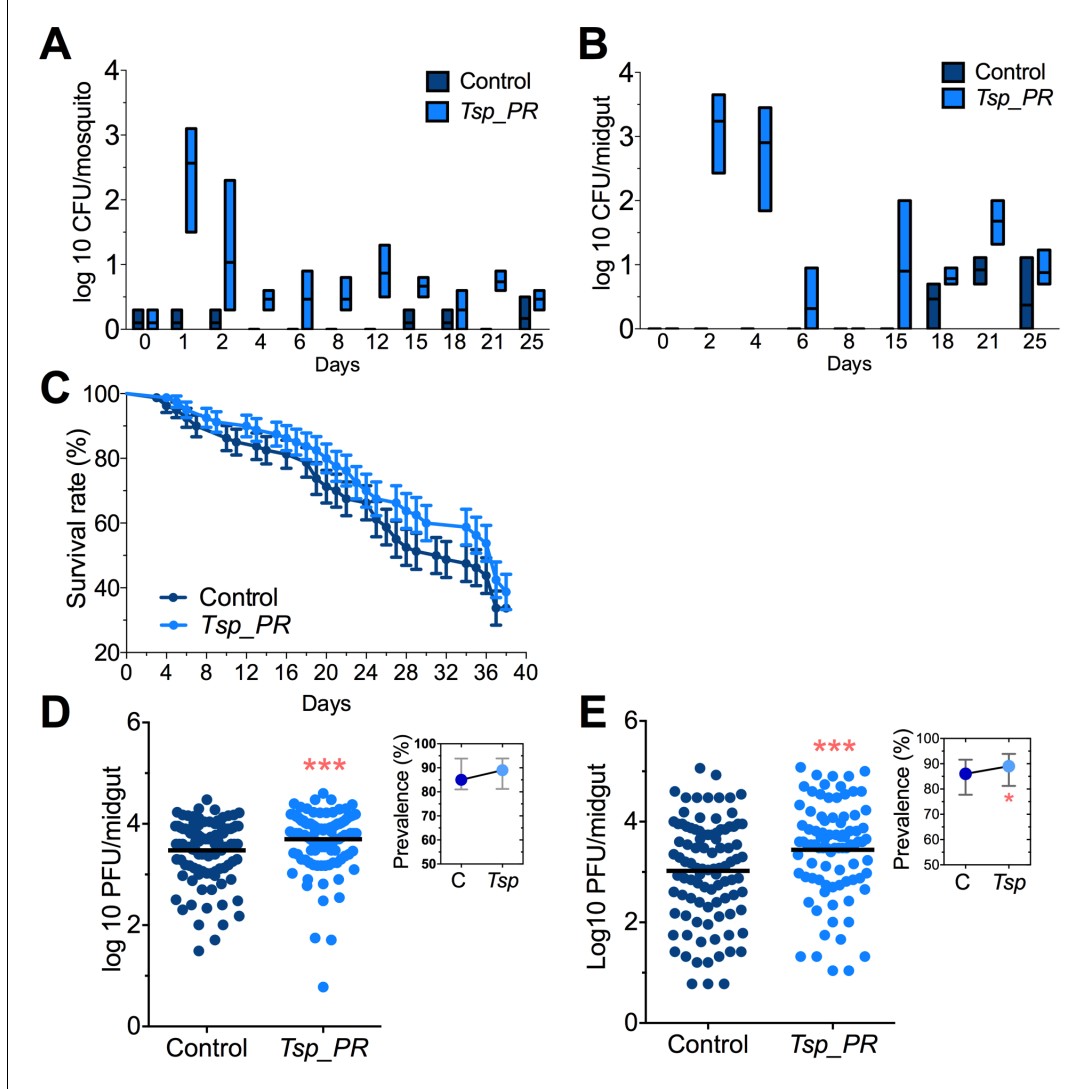

**Figure 2.** *Tsp_PR* fungus significantly increases DENV infection in *Aedes* mosquito midguts. *Aedes* mosquitoes were mock-fed or fed for 48 hr with 10% sucrose solution containing $1 \times 10^9$ *Tsp_PR* spores. After spore feeding, (**A**) Fungus colonization in whole mosquitoes or (**B**) midguts. The presence of *Tsp_PR* in the mosquito was monitored for 25 days after introduction by enumerating fungal CFUs on Sabouraud agar with antibiotics cocktail from three independent experiments, the line indicates the mean and bars the maximum and minimum ranges. (**C**) Survival assays. Female mosquitoes fed with *Tsp_PR* spores or unfed were monitored in a daily basis for 38 days in three independent experiments (N = 80, p=0.3073). Error bars represent ± SE. (**D**) Rockefeller strain mosquitoes, (Control, N = 123; *Tsp_PR*, N = 120) or (**E**) Orlando strain mosquitoes (Control, N = 113; *Tsp_PR*, N = 99) were infected with a blood meal containing DENV; at 7 days post-infection (dpi), the midguts were dissected. Each dot represents a plaque-forming unit (PFU) transformed to $\log_{10}$ in individual midguts from three independent experiments. The line indicates the mean. Upper right boxes show the prevalence of infected mosquitoes, error bars represent the 95% confidence interval. *p<0.05, ***p<0.001,.

DOI: https://doi.org/10.7554/eLife.28844.003

The following source data is available for figure 2:

**Source data 1.** Raw data and statistics summary for *Figure 2*.
DOI: https://doi.org/10.7554/eLife.28844.004

## *Tsp_PR*-secreted factors stimulate down-regulation of genes encoding blood-digesting enzymes

Next we explored the influence of the *Tsp_PR* secreted factors on the mosquito transcriptome, as a measure of its molecular physiology, in order to provide clues to the mechanism that could be responsible for the influence on DENV infection. A genome-wide microarray-based transcriptome comparison between the midguts of fungus-secretome solution-exposed and non-exposed

mosquitoes reveled regulation (23 up-regulated and 22 down-regulated) of a variety of genes belonging to different functional classes (*Table 1*) (*Figure 6A*). Forty-eight percent (11 genes) displaying increased transcript abundance after exposure to the *Tsp_PR* secretome belonged to the redox class and are putatively involved in various oxidoreductive processes, including detoxification. As many as 82% (9 genes) of this redox class encoded cytochrome P450 proteins, suggesting that the fungus activates the detoxification machinery in the mosquito midgut. A significant proportion (55%, 12 genes) of the 22 down-regulated genes were functionally related to blood digestion, and predominantly proteolysis (41%) (*Table 1*) (*Figure 6B*). Twenty-two percent of the total down-regulated genes encoded trypsins. These results suggest that the *Tsp_PR* secretome causes an impairment of the mosquito's ability to digest the blood meal.

## Fungus-secreted molecules impair trypsin activity of the *Ae. aegypti* midgut

Earlier studies have shown that mosquito ovary development is correlated with the amount of digested blood and reabsorption of nutrients for egg production (*Bryant et al., 2010*; *Lea et al., 1978*). Poor digestion of the blood meal therefore results in decreased ovary development. Hence, ovary development after blood feeding can serve as a proxy assay for blood-digestion efficiency. We performed an assay in which *Tsp_PR* secretome-exposed and subsequently blood-fed mosquitoes were microscopically evaluated for ovary follicle developmental stage after a non-infectious blood meal. Mosquitoes exposed for 48 hr to the fungus secretome prior to blood feeding showed a significantly decreased in ovary development (p=0.025). Only 9% of the mosquitoes had fully developed ovary follicles (*Figure 7A*, *Figure 7—source data 1*); in contrast, untreated mosquitoes had a higher level (29%) of fully ovary follicle development. We also, examined the mosquito body weight at 1 hr and 4 hr after a blood meal. Mosquitoes treated with *Tsp_PR* secretome were heavier than nontreated mosquitoes at 48 hr post-blood meal, indicating impairment of blood digestion (p=0.001) (*Figure 7B*, *Figure 7—source data 1*).

Following blood ingestion, gut epithelial cells synthesize and secrete diverse enzymes to digest the protein-rich meal (*Borovsky, 2003*). Trypsin activity accounts for most of the proteolytic activity during the mosquito's digestion of blood (*Barillas-Mury et al., 1995*; *Noriega and Wells, 1999*). Our transcriptomic analysis showed a down-regulation of several trypsin genes in fungus-treated mosquitoes in the absence of a blood meal.

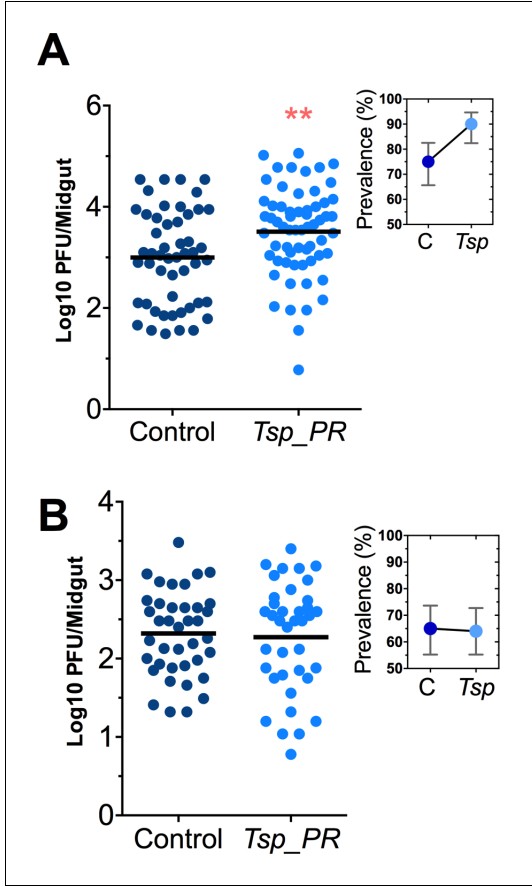

**Figure 3.** Heat-sensitive *Tsp_PR* secreted molecule(s) render mosquitoes more susceptible to DENV infection. DENV titers by plaque assay. Orlando strain mosquitoes were mock-fed or fed for 48 hr with a 10% sucrose solution a *Tsp_PR* filtered solution, which contained only (**A**) the fungus-secreted molecules (Control, N = 68; *Tsp_PR*, N = 68), or (**B**) a heat-treated *Tsp_PR* fungus-secreted molecules (Control, N = 60; *Tsp_PR*, N = 61). Mosquitoes were infected with a blood meal containing DENV, and midguts were dissected at 7 dpi. Each dot represents a $\log_{10}$ PFU in individual midguts from three independent experiments. The line indicates the mean. Upper right boxes show the prevalence of infected mosquitoes, error bars represent the 95% confidence interval. **p<0.01.

DOI: https://doi.org/10.7554/eLife.28844.005

The following source data is available for figure 3:

**Source data 1.** Raw data and statistics summary for *Figure 3*.
DOI: https://doi.org/10.7554/eLife.28844.006

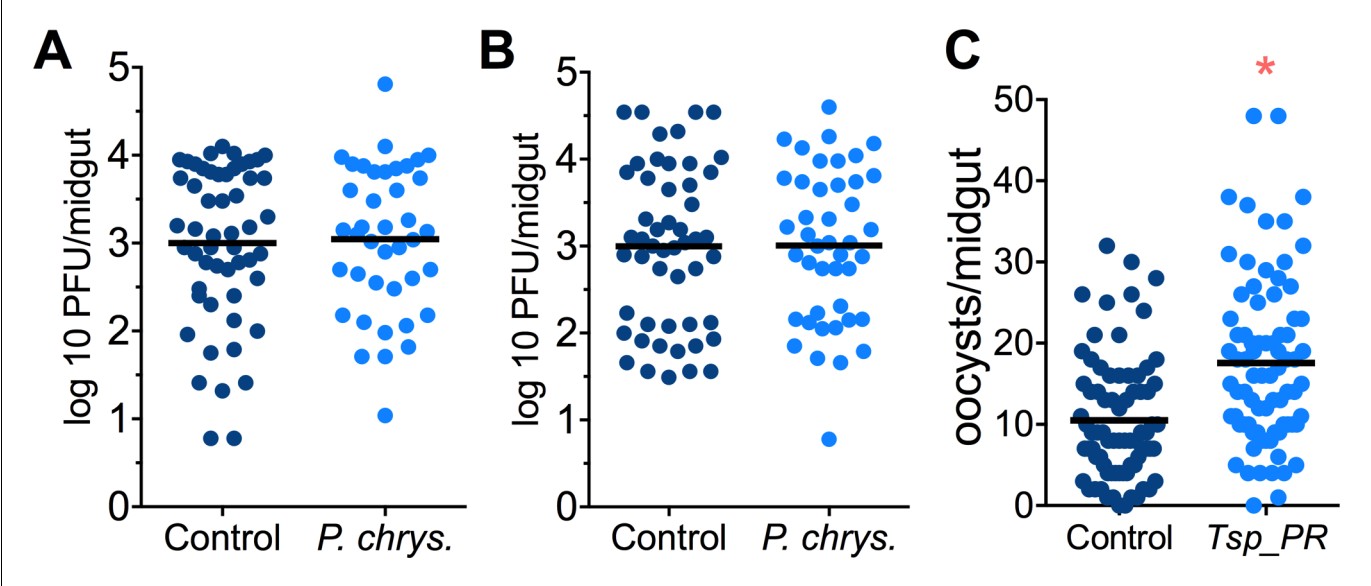

**Figure 4.** *Penicillium chrysogenum* does not modulate DENV infection in *Aedes* mosquito midguts, while *Tsp_PR* render *An. gambiae* more susceptible to *Plasmodium* infection. *Penicillium chrysogenum* was isolated from field-caught *Anopheles sp.* mosquitoes and re-introduced into *Aedes* mosquitoes to test the modulation of DENV infection. *Aedes* Orlando strain was mock-fed or fed for 48 hr with a 10% sucrose solution containing (**A**) $1 \times 10^9$ *P. chrysogenum* spores (Control, N = 61; *P. chrysogenum*, N = 47) or (**B**) fungus-secreted molecules (Control, N = 68; *P. chrysogenum*, N = 53). After fungus feeding, the mosquitoes were infected with a blood meal containing DENV; at 7 days post-infection (dpi), the midguts were dissected. Each dot represents a PFU value in individual midguts from three independent experiments. The line indicates the mean. (**C**) Influence of *Tsp_PR* on *P. falciparum* infection of *An. gambiae*, as measured by oocyst numbers 7 days after feeding on a *P. falciparum* gametocyte culture (infection intensity). The mosquito cohort (N = 79) that had been exposed to a *Tsp_PR* -laced sucrose solution for 48 hr prior to parasite infection had a significantly higher *P. falciparum* infection than did the non-fungus-exposed control cohort (N = 76). Graphs show three independent experiments. Each dot represents a single midgut, and bars represent the mean. *p<0.05.

DOI: https://doi.org/10.7554/eLife.28844.007

The following source data is available for figure 4:

**Source data 1.** Raw data and statistics summary for *Figure 4*.

DOI: https://doi.org/10.7554/eLife.28844.008

Next, we performed assays to determine whether the *Tsp_PR* secretome down-regulates midgut trypsin enzymatic activity in vivo, by simply assaying the enzymatic activity of mosquito guts after exposure to the *Tsp_PR* secretome at 24 hr after a blood meal. The results showed a significantly lower trypsin enzymatic activity in the midgut of the experimental *Tsp_PR* secretome-exposed group (p<0.001) than in the non-treated control mosquitoes (*Figure 7C*, *Figure 7—source data 1*).

The *Tsp_PR* secretome appeared to stimulate a reduction in trypsin enzymatic activity as a consequence of the transcriptional down-regulation of trypsin genes. However, we also proceeded to test whether the *Tsp_PR* secretome could directly influence trypsin enzymatic activity in vitro. This assay showed that the fungus secretome significantly decreased the in vitro activity of a commercial trypsin (p<0.01) in absence of mosquito's guts (*Figure 7D*, *Figure 7—source data 1*). To provide functional confirmation that the *Tsp_PR* secretome-regulated trypsins could influence DENV infection, we silenced selected trypsin genes using RNA interference (RNAi)-mediated gene silencing and compared DENV titers to that of a control cohort that has been treated with a GFP dsRNA. Silencing efficiency was evaluated (*Figure 8A*, *Figure 8—source data 1*) for AAEL010196, T196 (*Figure 8B*, *Figure 8—source data 1*); AAEL013707, T707 (*Figure 8C*, *Figure 8—source data 1*); AAEL013714, T714 (*Figure 8D*, *Figure 8—source data 1*); AAEL013715, T715 (*Figure 8E*, *Figure 8—source data 1*). Silencing of T714 and T715 resulted in a modest increased DENV infection intensity, while simultaneous silencing of all trypsin genes (Tmix) significantly increased DENV infection intensity (p<0.05). Silencing of T714 resulted in the greatest increase of DENV infection prevalence (p<0.001) (*Figure 8F*, *Figure 8—source data 1*). These results indicate that a transcript reduction of multiple

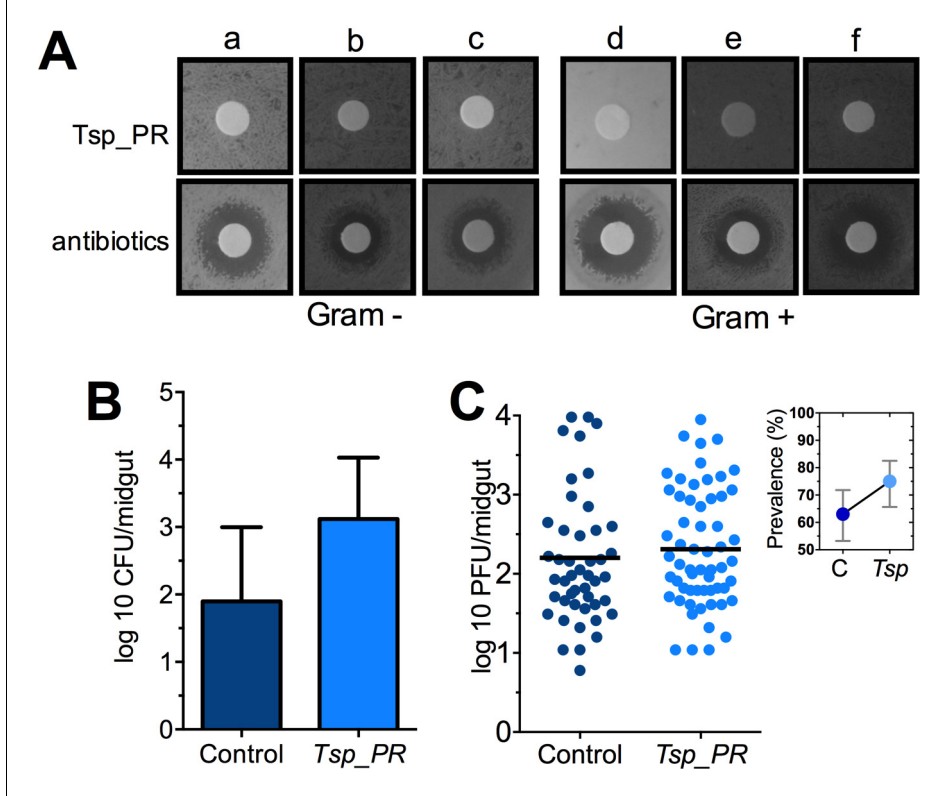

**Figure 5.** The *Tsp_PR* secreted molecule(s) do not affect bacterial load or DENV infection in aseptic mosquitoes. (**A**) Bacterial growth inhibition assay. Six bacterial isolates of field-caught mosquitoes (*Ramirez et al., 2012*) were independently plated on LB agar and covered with a disk soaked in a *Tsp_PR* secretome solution or antibiotic cocktail. Three isolates were Gram-negative bacteria: *Serratia marcescens* (**a**), *Chromobacterium haemolyticum* (**b**), and *Enterobacter hormaeche* (**c**). Three were Gram-positive bacteria: *Bacillus subtilis* (**d**), *Staphilococcus capprae* (**e**), and *Lactococcus lactis* (**f**). Bacterial inhibition was indicated by the presence of a bacterial inhibition zone around the disk. (**B**) Total bacterial loads. Midguts of secretome solution-exposed and unexposed mosquitoes were collected, homogenized, and plated on LB agar. Bacteria were counted as CFU. Error bars represent ± SD of three independent experiments p=0.202. (**C**) DENV titers in aseptic mosquitoes. Mosquitoes were treated with an antibiotic cocktail via a sugar meal 4 days before the fungal treatment and were mock-fed or fed for 48 hr on a *Tsp_PR* secretome solution prior to DENV infection. Each dot represents the PFU after 7 dpi in individual midguts from three independent experiments (Control, N = 75; *Tsp_PR*, N = 78). The line indicates the mean, p=0.867. Upper right box shows the prevalence of infected mosquitoes, error bars represent the 95% confidence interval.
DOI: https://doi.org/10.7554/eLife.28844.009

The following source data is available for figure 5:

**Source data 1.** Raw data and statistics summary for *Figure 5B,C*.
DOI: https://doi.org/10.7554/eLife.28844.010

---

trypsins has a promoting effect on DENV infection, and trypsin T714 appears to exert the strongest contribution.

Taken together, these results show that one or more *Tsp_PR*-secreted factors influence blood digestion through the modulation of trypsin expression and activity, which in turn, affects susceptibility to DENV infection.

## Discussion

Here we show that a fungus that is naturally associated with *Aedes* sp. in the field can enhance virus infection, and thereby potentially also enhance DENV transmission. Several studies have shown that the mosquito gut microbiota plays a critical role in determining the outcome of pathogen infection

**Table 1.** Significantly regulated genes in *Tsp_PR* secretome-exposed mosquitoes.
$Log_2$ values of differential mRNA abundances (*Tsp_PR* exposed/non-exposed) of genes.

| Gene description | Gene ID | $Log_2$ |
|---|---|---|
| trypsin | AAEL010196 | −2.33 |
| trypsin, putative | AAEL013714 | −2.25 |
| trypsin | AAEL010203 | −2.07 |
| Catalytic activity, serine-type endopeptidase activity, proteolysis | AAEL017520 | −1.99 |
| trypsin | AAEL013715 | −1.95 |
| serine-type enodopeptidase, putative | AAEL001690 | −1.64 |
| saccharopine dehydrogenase | AAEL014734 | −1.35 |
| Sialin, Sodium/sialic acid cotransporter, putative | AAEL004247 | −1.25 |
| hypothetical protein | AAEL013835 | −1.22 |
| alkaline phosphatase | AAEL000931 | −1.19 |
| trypsin | AAEL013707 | −1.19 |
| hypothetical protein | AAEL007591 | −1.08 |
| carboxypeptidase | AAEL010776 | −1.07 |
| triosephosphate isomerase | AAEL002542 | −1.03 |
| leucinech transmembrane proteins | AAEL005762 | −0.90 |
| serine-type enodopeptidase, putative | AAEL001701 | −0.90 |
| Conserved hypothetical protein (chitin-binding domain type 2) | AAEL017334 | −0.89 |
| sterol carrier protein-2, putative | AAEL012697 | −0.82 |
| hypothetical protein | AAEL002875 | −0.82 |
| lysosomal acid lipase, putative | AAEL004933 | −0.81 |
| hypothetical protein | AAEL002963 | −0.78 |
| lysosomal alpha-mannosidase (mannosidase alpha class 2b member) | AAEL005763 | −0.76 |
| ornithine decarboxylase | AAEL000044 | 1.70 |
| glucosyl/glucuronosyl transferases | AAEL003099 | 1.23 |
| conserved hypothetical protein(salivary protein [Culex]) | AAEL009985 | 1.05 |
| cytochrome P450 | AAEL014607 | 1.01 |
| cytochrome P450 | AAEL014609 | 0.99 |
| cytochrome P450 | AAEL006811 | 0.97 |
| cytochrome P450 | AAEL014616 | 0.96 |
| cytochrome P450 | AAEL014608 | 0.95 |
| hypothetical protein | AAEL004317 | 0.94 |
| hypothetical protein | AAEL005669 | 0.92 |
| hypothetical protein | AAEL002263 | 0.90 |
| glucosyl/glucuronosyl transferases | AAEL010386 | 0.88 |
| CRAL/TRIO domain-containing protein | AAEL003347 | 0.87 |
| alpha-amylase | AAEL010537 | 0.85 |
| hypothetical protein | AAEL011203 | 0.83 |
| glucose dehydrogenase | AAEL004027 | 0.82 |
| hypothetical protein | AAEL009198 | 0.81 |
| glutamate decarboxylase | AAEL010951 | 0.80 |
| cytochrome P450 | AAEL008846 | 0.79 |
| Vanin-like protein 1 precursor, putative | AAEL006023 | 0.78 |
| cytochrome b5, putative | AAEL012636 | 0.77 |

*Table 1 continued on next page*

*Table 1 continued*

| Gene description | Gene ID | Log$_2$ |
|---|---|---|
| cytochrome P450 | AAEL009131 | 0.76 |
| cytochrome P450 | AAEL014893 | 0.75 |

DOI: https://doi.org/10.7554/eLife.28844.012

and that different species of gut-associated bacteria can inhibit DENV and *Plasmodium* infection in *Aedes* and *Anopheles* mosquitoes, respectively (*Dong et al., 2009*; *Pumpuni et al., 1996*; *Ramirez et al., 2012*). Studies of the mosquito microbiota have mainly focused on bacteria, whereas the influence of fungi on mosquito physiology and pathogen infection remains largely understudied. Most studies concerning the mosquito mycobiota have explored the biology of entomopathogenic fungi, such as *Metarhizium anisopliae* and *Beauveria bassiana*, for use in mosquito control (*Darbro et al., 2011*; *Scholte et al., 2007*). The association of diverse fungi, with different developmental stages of mosquitoes in the field is reported (*da Costa and de Oliveira, 1998*; *da S Pereira et al., 2009*), and fungi, including *Talaromyces* sp., have also been isolated from the midgut of other arthropods such as ticks, sandflies, and kissing bugs (*Akhoundi et al., 2012*; *Marti et al., 2007*). Their presence is not surprising, since mosquitoes and many other insect vectors of disease are constantly exposed to a fungus-rich environment, including the larval breeding habitat, the plant nectars on which the adults feed, and the sites on which they rest.

We have shown that the presence of *Tsp_PR* spores or fungus secreted-molecules in the midgut tissue of the mosquito results in higher DENV infections in two different *Ae. aegypti* strains. However, modulation of infection does not seem to be a general property of a closely related *Penicillium* species, since exposure of *Ae. aegypti* to whole spores or the secretome of another mosquito-associated species, *Penicillium chrysogenum*, did not affect DENV infection (*Angleró-Rodríguez et al., 2016*). Our study suggests that *Tsp_PR* produces a factor(s) that is a likely a heat-sensitive protein or metabolite that alters the mosquito's susceptibility to DENV. Mosquitoes treated with the fungus secretome did not display differences in bacterial load in their midguts, indicating that the fungus-secreted factor(s) does not compromise the ability of *Ae. aegypti* to control its intestinal microbiota.

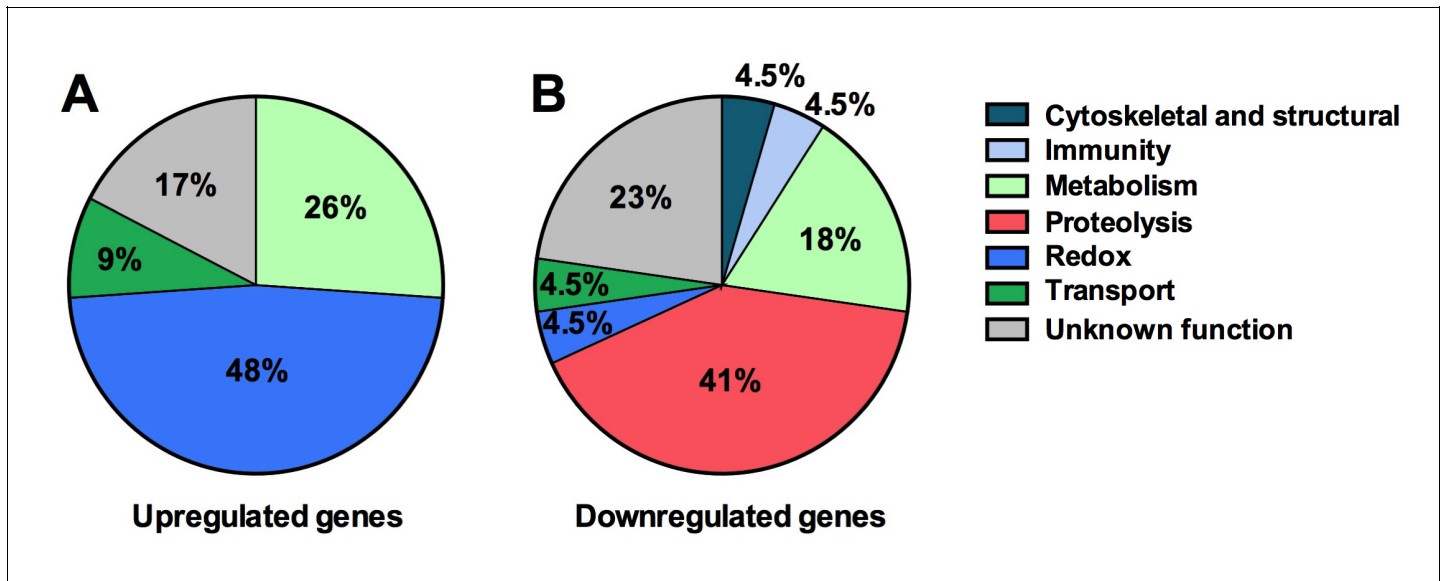

**Figure 6.** *Tsp_PR* secreted-molecule(s) –induced gene regulation. Functional classification in real numbers of the differentially expressed genes in mosquito midguts treated with *Tsp_PR* secretome for 48 hr, as compared to those of untreated mosquitoes. The fungus treatment-responsive genes are presented in *Table 1* and *Supplementary file 1*.
DOI: https://doi.org/10.7554/eLife.28844.011

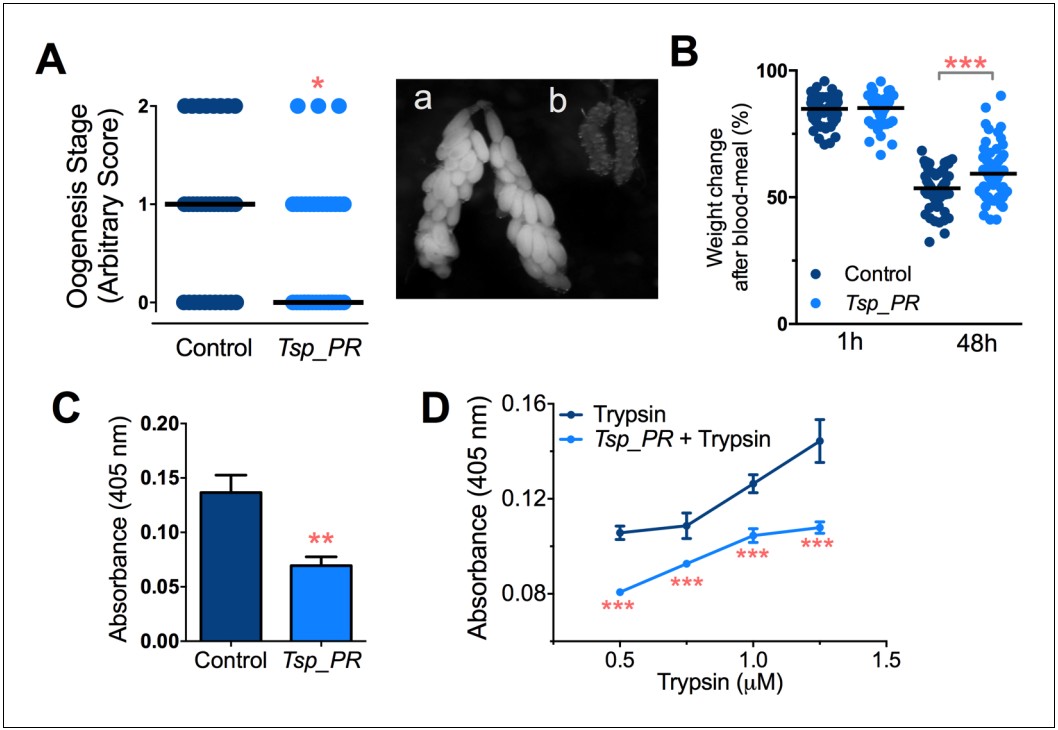

**Figure 7.** Inhibition of *Ae.aegypti* midgut trypsin activity by *Tsp_PR*-secreted molecule(s). (A) (*left*) Ovary development based on an arbitrary score of the ovary size at 6 days after a blood meal, 0 for small round follicles, 1 for intermediate size follicles, and 2 for fully developed follicles, with the elongated shape of normal mature eggs. Control, N = 34; *Tsp_PR*, N = 34, line represents the median, of three independent experiments. (*right*) Light microscopy picture of (a) a completely developed ovary follicle, which represents a score 2 (b) small round follicles, represent score 0. (B) Change in mosquito body weight after 1 hr (Control, N = 66; *Tsp_PR*, N = 74), (p=0.784) and 48 hr (Control, N = 58; *Tsp_PR*, N = 74) (p=0.001) of a non-infected blood meal (C) Trypsin in vivo enzymatic activity in midguts of mosquitoes treated or mock-treated with *Tsp_PR* secretome. Error bars represent ± SEM of three independent experiments. (D) Trypsin in vitro enzymatic assays of *Tsp_PR*'s ability to inhibit commercial trypsin activity. The activity was measured at various concentrations of trypsin. *Tsp_PR* represents the control group in which the fungus filtrate was added but no trypsin, and the absence of trypsin activity was experimentally confirmed (not shown). Error bars represent ± SEM of three independent experiments. *p<0.05, **p<0.01, ***p<0.01.

DOI: https://doi.org/10.7554/eLife.28844.013

The following source data is available for figure 7:

**Source data 1.** Raw data and statistics summary for *Figure 7*.

DOI: https://doi.org/10.7554/eLife.28844.014

---

Our transcriptomic analysis of how the *Tsp_PR* influences mosquito physiology pointed to a profound regulation of genes principally related to metabolism and digestion. Among the top up-regulated genes were ornithine decarboxylase, which is associated with polyamine biosynthesis; glucosyl/glucuronosyl transferases, which play an important role in the detoxification of xenobiotics and the regulation of endobiotics (*Ahn et al., 2012*); and cytochrome P450, which is involved in detoxification (*Strode et al., 2008*). Interestingly, a considerable number of up-regulated genes were associated with detoxification, suggesting that the fungus imposes a certain level of toxicity. However, *Tsp_PR* exposure did not seem to affect mosquito longevity. Perhaps the up-regulation of the detoxification machinery effectively reverses a detrimental effect.

Among the up-regulated genes, only glucosyl/glucuronosyl transferase (AAEL003099) modulates DENV infection in *Ae. aegypti* (*Sim et al., 2013*). An earlier study showed that this gene was expressed at a lower level in *Ae. aegypti* strains that are refractory to DENV and at a higher level in DENV-susceptible strains, suggesting that the AAEL003099 gene acts as a DENV host factor. These findings are in agreement with our study, in which *Tsp_PR* secretome-exposed mosquitoes showed

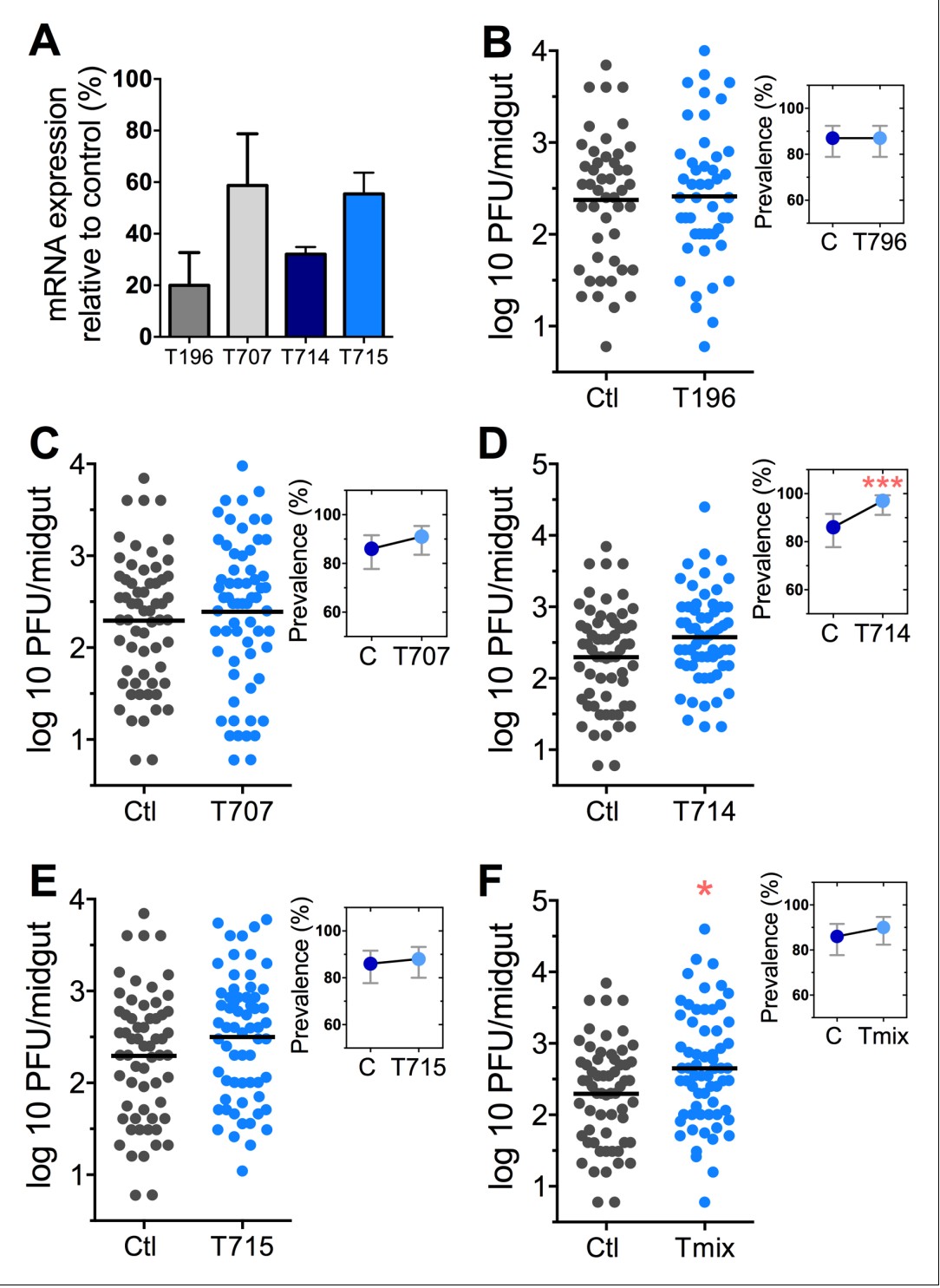

**Figure 8.** DENV infection after dsRNA-mediated silencing of trypsin genes. (A) Trypsin genes abundance after dsRNA-mediated gene silencing, (AAEL010196 (T196), AAEL013707 (T707), AAEL013714 (T714), AAEL013715 (T715). (B–F) DENV infection intensity of trypsin genes-silenced mosquitoes are compared to GFP dsRNA-treated control mosquitoes (B) T196 (Control, N = 54; T196, N = 54), (C) T707 (Control, N = 71; T707, N = 69), (D) T714 (Control, N = 71; T714, N = 62), (E) T715 (Control, N = 71; T715, N = 72), (F) Simultaneous silencing of all trypsins (Tmix) (Control, N = 71; Tmix, N = 71). The line indicates the mean, each dot represents the $\log_{10}$ PFU after 7 dpi in individual midguts from four independent biological experiments, except T196 which represents three

*Figure 8 continued on next page*

*Figure 8 continued*
independent experiments. Upper right boxes show the prevalence of infected mosquitoes, error bars represent the 95% confidence interval. *p<0.05, ***p<0.001.
DOI: https://doi.org/10.7554/eLife.28844.015
The following source data is available for figure 8:
**Source data 1.** Raw data and statistics summary for *Figure 8*.
DOI: https://doi.org/10.7554/eLife.28844.016

an elevated expression of AAEL003099. It is therefore possible that a *Tsp_PR*-mediated up-regulation of AAEL003099 contributes to the elevated DENV infection levels. Another up-regulated gene associated with proviral effect is ornithine decarboxylase, a recent study showed that the polyamine biosynthesis is associated with an enhanced infection of Chikungunya and Zika viruses in humans (*Mounce et al., 2016*).

The most striking expression signature that resulted from *Tsp_PR* secretome exposure was a profound overrepresentation of down-regulated proteolysis genes. Trypsin genes were highly represented in this category, and they play a crucial role in the blood-digesting process in the mosquito midgut. Two groups of trypsins are produced after the blood meal: early trypsins during the first 6 hr post-blood meal, and the late trypsins between 8 and 36 hr post-blood meal (*Barillas-Mury et al., 1995*; *Noriega and Wells, 1999*). However, the early trypsin mRNA is produced prior to feeding and is stored in the midgut epithelial cells, remaining untranslated until blood ingestion (*Noriega and Wells, 1999*). Our transcriptomic analysis revealed trypsin AAEL010196 to be the most down-regulated, followed by four other trypsin genes (AAEL013714, AAEL010203, AAEL013715, and AAEL013707). We confirmed the functional significance of trypsin transcript depletion on DENV infection using RNAi-mediated gene knock-down studies.

A phylogenetic analysis based on the trypsin gene nucleotide sequences revealed that the AAEL013715 and AAEL013707 cluster together and show a closer resemblance to the early trypsin EA1 gene, which has been associated with *Aedes* blood digestion (*Figure 9*). The trypsin genes AAEL013714, AAEL010203, and AAEL010196 have a close resemblance to two well-characterized late trypsins, 5G1 and LT (*Brackney et al., 2010*). Studies of trypsin 5G1 have established a link between trypsin activity and permissiveness to DENV infection, since RNAi-mediated repression of this gene results in greater susceptibility to the virus (*da Costa and de Oliveira, 1998*). This finding

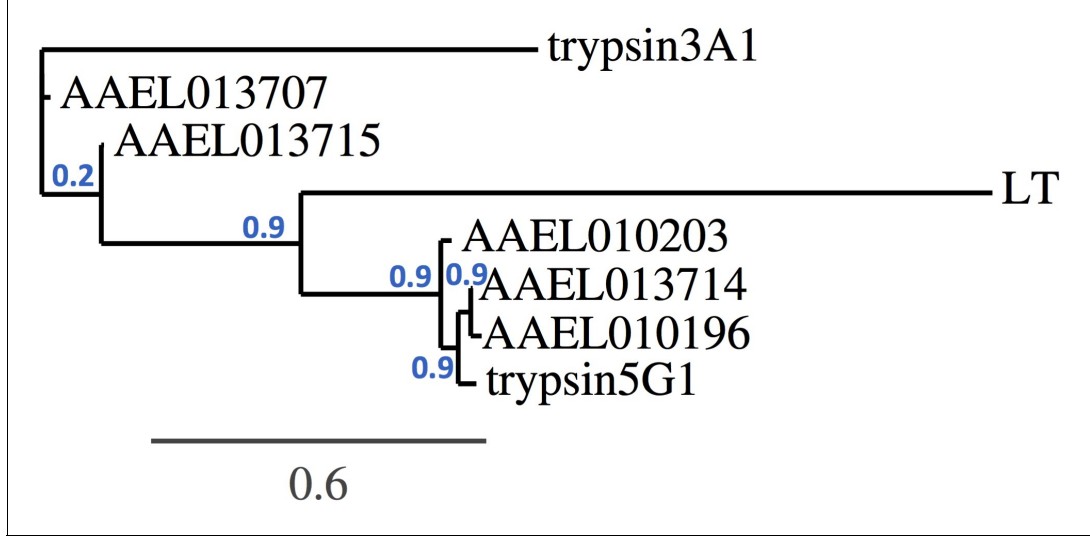

**Figure 9.** Trypsin phylogeny. Phylogenetic tree of the nucleotide alignment of trypsins regulated by *Tsp_PR* and others associated with the *Aedes* midgut. Branch support values represent approximate likelihood ratios, constructed using the program PhyML 3.0 approximate likelihood-ratio test (*Dereeper et al., 2008*).
DOI: https://doi.org/10.7554/eLife.28844.017

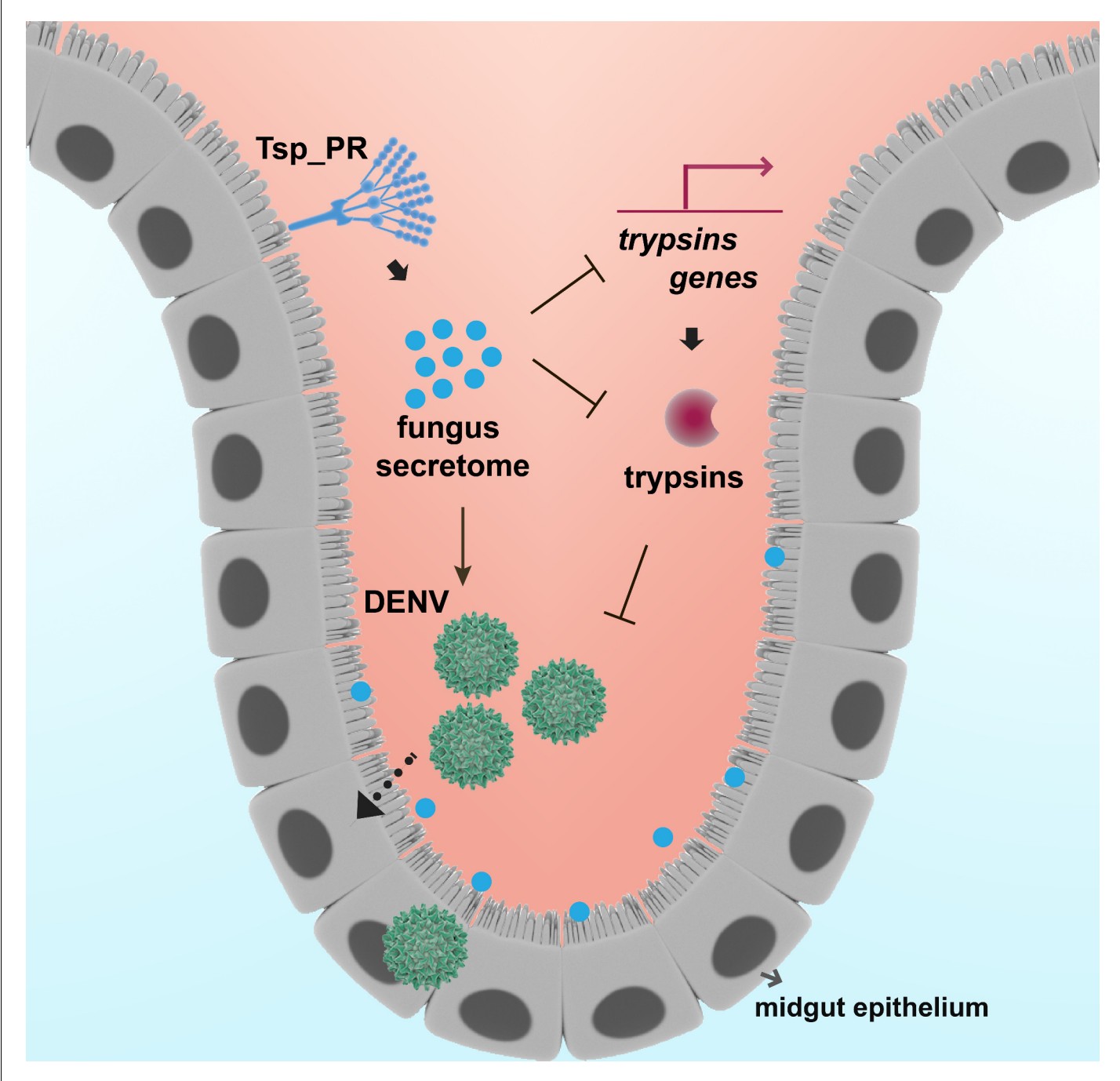

**Figure 10.** Model of *Tsp_PR*-mediated increased *Ae. aegypti* permissiveness to DENV. *Tsp_PR* secreted factors render *Ae. aegypti* more permissive to DENV through a mechanism that involves the down-regulation of gut trypsin transcripts and inhibition of enzymatic activity in the midgut epithelium. Trypsins have an antagonistic role in DENV infection. Decrease of trypsins abundance results in increased susceptible to DENV infection. Additional files.

DOI: https://doi.org/10.7554/eLife.28844.018

agrees with our studies showing that silencing of a closely related trypsin, the AAEL013714, also results in a greater susceptibility to DENV infection. QTL mapping of loci that determine permissiveness to DENV also indicated the likely involvement of trypsin genes (*Bosio et al., 2000*).

We have recently demonstrated that heat-stable factors secreted by a mosquito-associated *P. chrysogenum* fungus can render *Anopheles gambiae* mosquitoes more susceptible to the malaria parasite *Plasmodium* when present in the midgut tissue (*Angleró-Rodríguez et al., 2016*). We showed that this phenomenon was attributed to the up-regulation of an ornithine decarboxylase gene which in turn results in a suppression of nitric oxide –dependent parasite killing. While ingestion of *Tsp_PR* also resulted in a strong up-regulation of the *Ae. aegypti* ornithine decarboxylase gene, heat inactivation of its secreted factors abolished its ability to modulate DENV infection. Exposure of *An. gambiae* to *Tsp_PR* also rendered it more permissive to *Plasmodium* infection, but exposure of *Ae aegypti* to *P. chrysogenum* did not affect its permissiveness to DENV. Finally, exposure of *An. gambiae* to *P. chrysogenum* did not result in a down-regulation of blood digestive enzymes (*Angleró-Rodríguez et al., 2016*). These results indicate that that the two fungi influence infection with the virus and the parasite through different factors and mechanisms in their respective mosquito vectors. While both *Tsp_PR* and *P. chrysogenum* appears to secrete a factor that influences *An. gambiae* susceptibility to the malaria parasite, only *Tsp_PR* produces the factor that render *Ae. aegypti* more susceptible to DENV.

Our findings may have significant implications for the epidemiology and transmission of DENV by *Aedes* mosquitoes in the field. Fungi are abundantly associated with field mosquitoes (*da Costa and de Oliveira, 1998*; *da S Pereira et al., 2009*; *Marti et al., 2007*); our studies show that mosquitoes can acquire *Talaromyces* sp. fungus through sugar feeding and that it will successfully colonize the mosquito midgut for at least 25 days. Whether the exposure of mosquitoes to *Tsp_PR* in the field results in a greater permissiveness to DENV is unknown and difficult to assess, but greater permissiveness is most likely dependent on the intensity of the fungal exposure and the success of the fungus in persisting in the gut of various mosquito populations. A greater susceptibility of mosquitoes to DENV as a result of *Tsp_PR* exposure could translate into an enhanced transmission. However, this possibility remains to be addressed experimentally, perhaps by correlating the presence of *Tsp_PR* and DENV in field-caught mosquitoes in dengue-endemic areas. The observed impact of *Tsp_PR* on mosquito egg development in the ovary and the delayed degradation of the blood meal may suggest that exposure of mosquitoes to the amounts of the *Tsp_PR* secretome used in this study would impose a fitness cost.

In summary, we show how the mosquito mycobiota can influence *Ae. aegypti* vector competence for DENV by physiologically modulating midgut enzymes, one of the first barriers the pathogen encounters during infection (*Figure 10*). Our study represents a significant step toward the understanding of fungi-mosquito interactions and their possible implications for the transmission of DENV and perhaps other arboviruses.

## Materials and methods

### Key resources table

| Reagent type (species) or resource | Designation | Source or reference | Additional information |
|---|---|---|---|
| strain, strain background (*Aedes aegypti* Rockefeller strain) | Rock | other | From Johns Hopkins University |
| strain, strain background (*Aedes aegypti* Orlando strain) | Orl | other | From Johns Hopkins University |
| cell line (*Aedes albopictus* C6/36) | C6/36 | ATCC CRL-1660 | |
| cell line (Baby hamster kidney cells (BHK-21)) | BHK-21 | ATCC CCL-10 | |
| biological sample (*Talaromyces sp.*) | *Tsp_PR* | this paper | Collected from a wild-caught mosquito from Naguabo, Puerto Rico |
| biological sample (*Penicillium chrysogenum*) | *P. chrysogenum* | PMID 27678168 | |
| biological sample (Dengue virus 2 strain New Guinea C (NGC) | DENV | PMID 18604274 | |

*Continued on next page*

*Continued*

| Reagent type (species) or resource | Designation | Source or reference | Additional information |
|---|---|---|---|
| biological sample (*Plasmodium falciparum*) | *P. falciparum* | PMID 27678168 | From Johns Hopkins Malaria Research Institute |
| Low Input Quick Amp Labeling kit | | Agilent Technologies | |
| RNeasy Mini Kit | | QIAGEN | |
| MMLV Reverse Transcriptase kit | | Promega | |

## Ethics statement

This study was carried out in strict accordance with the recommendations in the Guide for the Care and Use of Laboratory Animals of the National Institutes of Health and the Animal Care and Use Committee of the Johns Hopkins University (Permit Number: M006H300). Mice were only used for mosquito rearing as a blood source, according to approved protocol. Commercial anonymous human blood was used for DENV infection assays in mosquitoes, and informed consent was therefore not applicable. The Johns Hopkins School of Public Health Ethics Committee approved this protocol.

## Cell culture and mosquito rearing

The *Ae. albopictus* cell line C6/36 (ATCC CRL-1660) was maintained in MEM (Gibco) supplemented with 10% FBS, 1% L-glutamine, 1% non-essential amino acids, and 1% penicillin/streptomycin. Baby hamster kidney cells (BHK-21, ATCC CCL-10) were maintained on DMEM (Gibco) supplemented with 10% FBS, 1% L-glutamine, 1% penicillin/streptomycin, and 5 µg/mL plasmocin (Invivogen, San Diego, CA). C6/36 cells and BHK-21 cells were incubated in 5% $CO_2$ at 32°C and 37°C, respectively. *Ae. aegypti* mosquitoes were maintained on a 10% sucrose solution at 27°C and 80% relative humidity with a 14:10 hr light:dark cycle.

## Fungus treatments

*Tsp_PR* was grown on Sabouraud glucose agar (SGA) and identified as described in (*Angleró-Rodríguez et al., 2016*); spores were collected in PBS, counted, and resuspended in a 10% sucrose solution containing $1 \times 10^9$ spores. For fungal filtrate, spores were collected in a 10% sucrose solution and kept in a rocker machine overnight at 4°C. The next day, the solution was centrifuged at 470 rcf to collect the supernatant, which was passed through a 0.2-micron filter to remove fungus mycelia and spores. The heat-inactivated filtrate was treated in the same way, but the supernatant was incubated for 2 hr at 95°C. Adult female *Ae. aegypti* (3–4 days old) were starved for 6 hr and fed with the appropriate fungal treatment for 48 hr.

## DENV infection

DENV2 strain New Guinea C (NGC) was propagated in C6/36 cells, and titers were determined on BHK-21 cells by plaque assay. Mosquito infections with DENV were carried out as previously described (*Das et al., 2007*). In brief, DENV2-NGC was propagated in C6/36 cells for 6 days. Virus suspension was mixed 1:1 with commercial human blood and supplemented with 10% human serum and 100 µM ATP. Mosquitoes were infected via an artificial membrane feeder at 37°C for 30 min. Midguts were dissected and individually collected at 7 days post-infection.

## *Plasmodium* infection and oocyst enumeration

*Plasmodium falciparum* infections were performed following a standard protocol (*Dong et al., 2006*). At 48 hr post-feeding on fungi or filtrate, mosquitoes were fed on an NF54W strain gametocyte culture mixed with human blood, through a membrane feeder at 37°C. Engorged mosquitoes were maintained at 27°C for up to 8 days. *P. falciparum*-infected mosquito midguts were dissected and stained with 0.1% mercurochrome, and oocyst numbers were determined using a light microscope.

## Plaque assay

BHK-21 cells were seeded in 24-well plates the day before the assay. The next day, individual midguts were homogenized in DMEM with 0.5 mm glass beads using a Bullet Blender (NextAdvance). Homogenates were centrifuged at 18,400 rcf and the virus-containing suspensions were 10-fold serially diluted, and 100 uL of each dilution were inoculated onto 80% confluent BHK-21 plates. Plates were rocked for 15 min at room temperature and then incubated for 45 min at 37°C and 5% $CO_2$. After the incubation, 1 mL of DMEM containing 2% FBS and 0.8% methylcellulose was added to each well, and plates were incubated for 5 days. Plates were fixed and stained for 30 min with a 1:1 methanol/acetone and 1% crystal violet mixture. Then, plates were washed with water and the plaque-forming units counted.

## Microbial enumeration

Mosquitoes were surfaced-sterilized for 1 min in 70% ethanol and rinsed twice with 1X PBS. Mosquitoes were dissected, and five midguts were pooled in a microcentrifuge tube containing 150 µl of sterile PBS. Midguts were homogenized with a pestle and plated on LB agar for bacterial enumeration or Sabouraud agar with an antibiotic cocktail of penicillin/streptomycin and gentamicin for fungal enumeration (whole mosquitoes or midguts were collected for this procedure). Plates were incubated at room temperature for 48 hr for bacteria and 4 days for fungus, and then plates were counted as colony-forming units (CFU). *Bacterial growth inhibition assay*. Was performed using a disc diffusion test, in which a sterile filter paper disk soaked in a fungus secretome solution or an antibiotic cocktail as a control, was placed over a bacterial culture on LB agar medium. Plates were incubated for 24 hr at 32°C and the inhibition zone were evaluated.

## Aseptic mosquitoes

Adult female mosquitoes were maintained on 10% sucrose solution containing 75 µg/mL gentamicin sulfate and 100 units (µg)/mL of penicillin-streptomycin. Treatment was carried out for 4 days. To validate the efficiency of antibiotic treatment, midguts from control untreated and antibiotic treated mosquitoes were subjected to CFU assays. Mosquitoes were treated with the bacteria-free fungus secretome for 2 days, and then maintained on antibiotic-treated sucrose after the DENV infection.

## Genome-wide microarray-based transcriptome profiling

Transcriptome assays were conducted and analyzed as reported previously with a custom-designed full genome *Ae. aegypti* Agilent-based microarray platform (*Sim et al., 2013*; *Xi et al., 2008*). In brief, 200 ng of total RNA per replicate was used to synthesize Cy3 or Cy5-labeled cRNA probes using a Low Input Quick Amp Labeling kit (Agilent Technologies). Probes from midguts of *Tsp_PR*-treated mosquitoes were individually hybridized against probes from untreated mosquitoes as a control. The arrays were scanned with an Agilent Scanner. Transcription data were processed by beginning with background subtraction of the median fluorescent values, normalized with the LOWESS normalization method. Cy5/Cy3 ratios from replicate assays were subjected to *t*-tests at a significance level of $p < 0.05$ using TIGR MIDAS and MeV software. Transcript abundance data from all replicate assays were averaged with the GEPAS microarray preprocessing software and transformed to a logarithm (base 2). Self-self hybridizations have previously been used to determine a cutoff value for the significance of gene regulation on these microarrays of 0.75 in $\log_2$ scale (*Yang et al., 2002*). Three independent experiments were performed. Numeric microarray gene expression data are presented in *Table 1*, *Supplementary file 1*.

## Oogenesis assays

Mosquitoes we exposed or not exposed to the *Tsp_PR* secretome for 48 hr, after they had received a blood meal. Fully engorged females from both groups were collected and maintained for 6 days; their ovaries were then dissected in PBS, and oogenesis was microscopically evaluated. Through microscopic evaluation of the follicle development in the ovary, we assigned an arbitrary score of 0 for small round follicles, 1 for intermediate size follicles, and 2 for fully developed follicles, with the elongated shape of normal mature eggs.

## Body weight measurements

Mosquitoes were fed with non-infectious blood for 20 min, then fully-engorged mosquitoes were individually collected and placed in individual round-bottom conical tubes and incubated without sugar or water. The conical tube weight was measured prior to collecting the mosquito. The weight of mosquitoes at 1 hr post-infection was calculated by subtracting the empty tube weight from the total weight. At 48 hr mosquitoes were cold anesthetized and weighted directly to the analytical balance without the tube to avoid accumulated excreta.

## Trypsin activity assay

### Mosquito endogenous trypsin activity assay

Mosquitoes were exposed, or not exposed, to the *Tsp_PR* secretome for 48 hr, after they were fed an artificial blood meal (40% PBS, 50% FBS, 1 mM ATP, and 2 mg of phenol red) to avoid interference with the assay (*Brackney et al., 2008*). At 24 hr post-blood meal, 10 midguts per group were dissected, collected in 50 μL of buffer solution (50 mM Tris-HCl, pH 8.0, with 10 mM $CaCl_2$), and homogenized on ice with a pestle. Supernatants were collected after a high-speed centrifugation (18,400 rcf) at 4°C. Trypsin activity assays were performed using the synthetic colorimetric substrate $N_\alpha$-benzoyl-D,L-arginine-p-nitroanilide hydrochloride (BApNA) (Sigma). The reaction mixtures, each containing 5 μl of midgut extract and 1 mM BApNA, were then incubated at 37°C for 5 min. Absorbance values were measured in a plate reader at A405 nm.

### In vitro trypsin inhibition assay

A reaction mixture was made using equal volumes of *Tsp_PR* filtrate and commercial trypsin (105 μM) (Gibco). The mixture was incubated at 27°C for 3 hr. The reaction solution was prepared by adding increasing volumes of the 1:1 mixture to the buffer solution and 1 mM BApNA, to a final volume of 200 uL. Absorbance was measured as described above.

## dsRNA-mediated gene silencing

The *trypsin* genes were depleted from adult female mosquitoes using established RNA interference (RNAi) methodology (*Sim et al., 2013*). Mosquitoes injected with *GFP* dsRNA were used as a control and RNAi assays were repeated three times. Gene silencing was verified by qRT-PCR at 3 days post-injection using RNA extracted from five whole mosquitoes per independent experiment. The primers to produce PCR Amplicons for dsRNA synthesis and qRT-PCR are presented in *Supplementary file 2*.

## Quantitative RT-PCR

Mosquito samples were collected in RLT buffer (QIAGEN), and then stored at −80°C until extraction. Total RNA was extracted using the RNeasy Mini Kit (QIAGEN); samples were treated with Turbo DNase (Ambion) before reverse transcription with a MMLV Reverse Transcriptase kit (Promega) according to the manufacturer's instructions. The cDNA was then used to determine gene expression by quantitative PCR using SYBR Green PCR Master Mix (Applied Biosystem). The transcript abundance of trypsin was compared to the expression of the ribosomal protein gene S7 as a normalization control. qPCR primers were designed to amplify the mRNA transcript outside the dsRNA region. However, groups where trypsins were silenced simultaneously the qPCR primers for a given trypsin crossed detected the injected dsRNA of other highly similar trypsins making unable the evaluation of silencing efficiency.

## Statistical analysis

To compare DENV titers between groups, *P*-values were calculated using Generalized Linear Regression (GLM) with experiment-clustered robust variance estimates to account for potential within-experiment correlation of outcomes (*Rogers, 1993*). The models included different link functions for various outcomes: identity link was used to compare DENV PFU and bacteria CFU, log-link with Poisson distribution was used for modeling infections and score 0 to 2 for oogenesis assays and log-link with Negative Binomial distribution was used to model number of *Plasmodium falciparum* oocysts. The models included an indicator variable for treatment with only two levels or multiple indicator variables for multiple treatment groups. Wald test *P*-values are reported in the results. Survival was

analyzed using the Log-rank (Mantel-Cox) test in Graphpad Prism. See source data file for the summary of the statistics.

## Acknowledgements

We would like to thank the Johns Hopkins Malaria Research Institute Insectary and the Microarray Core Facilities. We also thank Dr. Deborah McClellan for editing the manuscript. We thank for the support with the statistical analysis from the National Center for Research Resources and the National Center for Advancing Translational Sciences (NCATS) of the National Institutes of Health through Grant Number 1UL1TR001079. This work has been supported by National Institutes of Health, National Institute of Allergy and Infectious Diseases grant R01AI101431. We thank the Bloomberg Philanthropies for their support.

## Additional information

### Funding

| Funder | Grant reference number | Author |
| --- | --- | --- |
| National Institute of Allergy and Infectious Diseases | RO1AI101431 | Yesseinia I Angleró-Rodríguez Jenny Carlson George Dimopoulos |

The funders had no role in study design, data collection and interpretation, or the decision to submit the work for publication.

### Author contributions

Yesseinia I Angleró-Rodríguez, Conceptualization, Data curation, Formal analysis, Validation, Investigation, Visualization, Methodology, Writing—original draft, Writing—review and editing; Octavio AC Talyuli, Benjamin J Blumberg, Seokyoung Kang, Celia Demby, Alicia Shields, Jenny Carlson, Natapong Jupatanakul, Investigation, Methodology; George Dimopoulos, Conceptualization, Resources, Supervision, Funding acquisition, Project administration, Writing—review and editing

### Author ORCIDs

George Dimopoulos http://orcid.org/0000-0001-6755-8111

### Ethics

Animal experimentation: This study was carried out in strict accordance with the recommendations in the Guide for the Care and Use of Laboratory Animals of the National Institutes of Health and the Animal Care and Use Committee of the Johns Hopkins University (Permit Number: M006H300). Mice were only used for mosquito rearing as a blood source, according to approved protocol.

### Decision letter and Author response

Decision letter https://doi.org/10.7554/eLife.28844.022
Author response https://doi.org/10.7554/eLife.28844.023

## Additional files

### Supplementary files

• Supplementary file 1. Table shows all the genes that had differential mRNA abundances (Tsp_PR exposed/non-exposed) over or under the significance cutoff value of $\pm0.75$ Log$_2$.
DOI: https://doi.org/10.7554/eLife.28844.019

• Supplementary file 2. Primer sequences used for dsRNA synthesis and qPCR. Sequences underlined corresponds to T7 promoter
DOI: https://doi.org/10.7554/eLife.28844.020

• Transparent reporting form

DOI: https://doi.org/10.7554/eLife.28844.021

# Refernces

**Ahn SJ**, Vogel H, Heckel DG. 2012. Comparative analysis of the UDP-glycosyltransferase multigene family in insects. *Insect Biochemistry and Molecular Biology* **42**:133–147. DOI: https://doi.org/10.1016/j.ibmb.2011.11.006, PMID: 22155036

**Akhoundi M**, Bakhtiari R, Guillard T, Baghaei A, Tolouei R, Sereno D, Toubas D, Depaquit J, Abyaneh MR. 2012. Diversity of the bacterial and fungal microflora from the midgut and cuticle of phlebotomine sand flies collected in North-Western Iran. *PLoS One* **7**:e50259. DOI: https://doi.org/10.1371/journal.pone.0050259, PMID: 23226255

**Angleró-Rodríguez YI**, Blumberg BJ, Dong Y, Sandiford SL, Pike A, Clayton AM, Dimopoulos G. 2016. A natural Anopheles-associated Penicillium chrysogenum enhances mosquito susceptibility to Plasmodium infection. *Scientific Reports* **6**:34084. DOI: https://doi.org/10.1038/srep34084, PMID: 27678168

**Bahia AC**, Dong Y, Blumberg BJ, Mlambo G, Tripathi A, BenMarzouk-Hidalgo OJ, Chandra R, Dimopoulos G. 2014. Exploring Anopheles gut bacteria for Plasmodium blocking activity. *Environmental Microbiology* **16**: 2980–2994. DOI: https://doi.org/10.1111/1462-2920.12381, PMID: 24428613

**Bara R**, Zerfass I, Aly AH, Goldbach-Gecke H, Raghavan V, Sass P, Mándi A, Wray V, Polavarapu PL, Pretsch A, Lin W, Kurtán T, Debbab A, Brötz-Oesterhelt H, Proksch P. 2013. Atropisomeric dihydroanthracenones as inhibitors of multiresistant Staphylococcus aureus. *Journal of Medicinal Chemistry* **56**:3257–3272. DOI: https://doi.org/10.1021/jm301816a, PMID: 23534483

**Barillas-Mury CV**, Noriega FG, Wells MA. 1995. Early trypsin activity is part of the signal transduction system that activates transcription of the late trypsin gene in the midgut of the mosquito, Aedes aegypti. *Insect Biochemistry and Molecular Biology* **25**:241–246. DOI: https://doi.org/10.1016/0965-1748(94)00061-L, PMID: 7711754

**Borovsky D**. 2003. Biosynthesis and control of mosquito gut proteases. *IUBMB Life* **55**:435–441. DOI: https://doi.org/10.1080/15216540310001597721, PMID: 14609198

**Bosio CF**, Fulton RE, Salasek ML, Beaty BJ, Black WC. 2000. Quantitative trait loci that control vector competence for dengue-2 virus in the mosquito Aedes aegypti. *Genetics* **156**:687–698. PMID: 11014816

**Brackney DE**, Foy BD, Olson KE. 2008. The effects of midgut serine proteases on dengue virus type 2 infectivity of Aedes aegypti. *The American Journal of Tropical Medicine and Hygiene* **79**:267–274. PMID: 18689635

**Brackney DE**, Isoe J, W C B, Zamora J, Foy BD, Miesfeld RL, Olson KE. 2010. Expression profiling and comparative analyses of seven midgut serine proteases from the yellow fever mosquito, *Aedes aegypti*. *Journal of Insect Physiology* **56**:736–744. DOI: https://doi.org/10.1016/j.jinsphys.2010.01.003, PMID: 20100490

**Bryant B**, Macdonald W, Raikhel AS. 2010. microRNA miR-275 is indispensable for blood digestion and egg development in the mosquito Aedes aegypti. *PNAS* **107**:22391–22398. DOI: https://doi.org/10.1073/pnas.1016230107, PMID: 21115818

**da Costa GL**, de Oliveira PC. 1998. Penicillium species in mosquitoes from two Brazilian regions. *Journal of Basic Microbiology* **38**:343–347. DOI: https://doi.org/10.1002/(SICI)1521-4028(199811)38:5/6<343::AID-JOBM343>3.0.CO;2-Z, PMID: 9871332

**da S Pereira E**, de M Sarquis MI, Ferreira-Keppler RL, Hamada N, Alencar YB. 2009. Filamentous fungi associated with mosquito larvae (Diptera: Culicidae) in municipalities of the brazilian amazon. *Neotropical Entomology* **38**:352–359. DOI: https://doi.org/10.1590/S1519-566X2009000300009, PMID: 19618051

**Darbro JM**, Graham RI, Kay BH, Ryan PA, Thomas MB. 2011. Evaluation of entomopathogenic fungi as potential biological control agents of the dengue mosquito, *Aedes aegypti* (Diptera: Culicidae) . *Biocontrol Science and Technology* **21**:1027–1047. DOI: https://doi.org/10.1080/09583157.2011.597913

**Das S**, Garver L, Ramirez JR, Xi Z, Dimopoulos G. 2007. Protocol for dengue infections in mosquitoes (A. aegypti) and infection phenotype determination. *Journal of Visualized Experiments* **220**. DOI: https://doi.org/10.3791/220

**Dennison NJ**, Jupatanakul N, Dimopoulos G. 2014. The mosquito microbiota influences vector competence for human pathogens. *Current Opinion in Insect Science* **3**:6–13. DOI: https://doi.org/10.1016/j.cois.2014.07.004

**Dereeper A**, Guignon V, Blanc G, Audic S, Buffet S, Chevenet F, Dufayard JF, Guindon S, Lefort V, Lescot M, Claverie JM, Gascuel O. 2008. Phylogeny.fr: robust phylogenetic analysis for the non-specialist. *Nucleic Acids Research* **36**:W465–W469. DOI: https://doi.org/10.1093/nar/gkn180, PMID: 18424797

**Dong Y**, Aguilar R, Xi Z, Warr E, Mongin E, Dimopoulos G. 2006. Anopheles gambiae immune responses to human and rodent Plasmodium parasite species. *PLoS Pathogens* **2**:e52. DOI: https://doi.org/10.1371/journal.ppat.0020052, PMID: 16789837

**Dong Y**, Manfredini F, Dimopoulos G. 2009. Implication of the mosquito midgut microbiota in the defense against malaria parasites. *PLoS Pathogens* **5**:e1000423. DOI: https://doi.org/10.1371/journal.ppat.1000423, PMID: 19424427

**Dong Y**, Morton JC, Ramirez JL, Souza-Neto JA, Dimopoulos G. 2012. The entomopathogenic fungus Beauveria bassiana activate toll and JAK-STAT pathway-controlled effector genes and anti-dengue activity in Aedes aegypti. *Insect Biochemistry and Molecular Biology* **42**:126–132. DOI: https://doi.org/10.1016/j.ibmb.2011.11.005, PMID: 22198333

Jaber S, Mercier A, Knio K, Brun S, Kambris Z. 2016. Isolation of fungi from dead arthropods and identification of a new mosquito natural pathogen. *Parasites & Vectors* **9**:491. DOI: https://doi.org/10.1186/s13071-016-1763-3, PMID: 27595597

Klitgaard A, Iversen A, Andersen MR, Larsen TO, Frisvad JC, Nielsen KF. 2014. Aggressive dereplication using UHPLC-DAD-QTOF: screening extracts for up to 3000 fungal secondary metabolites. *Analytical and Bioanalytical Chemistry* **406**:1933–1943. DOI: https://doi.org/10.1007/s00216-013-7582-x, PMID: 24442010

Lea AO, Briegel H, Lea HM. 1978. Arrest, resorption, or maturation of oöcytes in Aedes aegypti: dependence on the quantity of blood and the interval between blood meals. *Physiological Entomology* **3**:309–316. DOI: https://doi.org/10.1111/j.1365-3032.1978.tb00164.x

Marti GA, García JJ, Cazau MC, López Lastra CC. 2007. Fungal flora of the digestive tract of Triatoma infestans (Hemiptera: Reduviidae) from Argentina. *Boletín De La Sociedad Argentina De Botánica* **42**:175–179 .

Molina-Cruz A, Gupta L, Richardson J, Bennett K, Black W, Barillas-Mury C. 2005. Effect of mosquito midgut trypsin activity on dengue-2 virus infection and dissemination in Aedes aegypti. *The American Journal of Tropical Medicine and Hygiene* **72**:631–637. PMID: 15891140

Mounce BC, Poirier EZ, Passoni G, Simon-Loriere E, Cesaro T, Prot M, Stapleford KA, Moratorio G, Sakuntabhai A, Levraud JP, Vignuzzi M. 2016. Interferon-Induced spermidine-spermine acetyltransferase and polyamine depletion restrict zika and chikungunya viruses. *Cell Host & Microbe* **20**:167–177. DOI: https://doi.org/10.1016/j.chom.2016.06.011, PMID: 27427208

Noriega FG, Wells MA. 1999. A molecular view of trypsin synthesis in the midgut of Aedes aegypti. *Journal of Insect Physiology* **45**:613–620. DOI: https://doi.org/10.1016/S0022-1910(99)00052-9, PMID: 12770346

Pitt J. 2014. Penicillium and Talaromyces: Introduction. In: Carl B, Tortorelo M (Eds). *Encyclopedia of Food Microbiology*. p. 6–7.

Pumpuni CB, Demaio J, Kent M, Davis JR, Beier JC. 1996. Bacterial population dynamics in three anopheline species: the impact on Plasmodium sporogonic development. *The American Journal of Tropical Medicine and Hygiene* **54**:214–218. DOI: https://doi.org/10.4269/ajtmh.1996.54.214, PMID: 8619451

Ramirez JL, Short SM, Bahia AC, Saraiva RG, Dong Y, Kang S, Tripathi A, Mlambo G, Dimopoulos G. 2014. Chromobacterium Csp_P reduces malaria and dengue infection in vector mosquitoes and has entomopathogenic and in vitro anti-pathogen activities. *PLoS Pathogens* **10**:e1004398. DOI: https://doi.org/10.1371/journal.ppat.1004398, PMID: 25340821

Ramirez JL, Souza-Neto J, Torres Cosme R, Rovira J, Ortiz A, Pascale JM, Dimopoulos G. 2012. Reciprocal tripartite interactions between the Aedes aegypti midgut microbiota, innate immune system and dengue virus influences vector competence. *PLoS Neglected Tropical Diseases* **6**:e1561. DOI: https://doi.org/10.1371/journal.pntd.0001561, PMID: 22413032

Rogers WH. 1993. Regression standard errors in clustered samples. *Stata Technical Bulletin* **13**:19–23.

Scholte EJ, Takken W, Knols BG. 2007. Infection of adult Aedes aegypti and Ae. albopictus mosquitoes with the entomopathogenic fungus Metarhizium anisopliae. *Acta Tropica* **102**:151–158. DOI: https://doi.org/10.1016/j.actatropica.2007.04.011, PMID: 17544354

Sim S, Jupatanakul N, Ramirez JL, Kang S, Romero-Vivas CM, Mohammed H, Dimopoulos G. 2013. Transcriptomic profiling of diverse Aedes aegypti strains reveals increased basal-level immune activation in dengue virus-refractory populations and identifies novel virus-vector molecular interactions. *PLoS Neglected Tropical Diseases* **7**:e2295. DOI: https://doi.org/10.1371/journal.pntd.0002295, PMID: 23861987

Strode C, Wondji CS, David JP, Hawkes NJ, Lumjuan N, Nelson DR, Drane DR, Karunaratne SH, Hemingway J, Black WC, Ranson H. 2008. Genomic analysis of detoxification genes in the mosquito Aedes aegypti. *Insect Biochemistry and Molecular Biology* **38**:113–123. DOI: https://doi.org/10.1016/j.ibmb.2007.09.007, PMID: 18070670

Xi Z, Ramirez JL, Dimopoulos G. 2008. The Aedes aegypti toll pathway controls dengue virus infection. *PLoS Pathogens* **4**:e1000098. DOI: https://doi.org/10.1371/journal.ppat.1000098, PMID: 18604274

Yang IV, Chen E, Hasseman JP, Liang W, Frank BC, Wang S, Sharov V, Saeed AI, White J, Li J, Lee NH, Yeatman TJ, Quackenbush J. 2002. Within the fold: assessing differential expression measures and reproducibility in microarray assays. *Genome Biology* **3**:research0062. DOI: https://doi.org/10.1186/gb-2002-3-11-research0062, PMID: 12429061

