## [Decision Letter]

[Editors’ note: a previous version of this study was rejected after peer review, but the authors submitted for reconsideration. The first decision letter after peer review is shown below.]

Thank you for submitting your work entitled "An *Aedes aegypti*-associated fungus increases susceptibility to dengue virus by modulating gut digestive activity" for consideration by *eLife*. Your article has been reviewed by three peer reviewers, and the evaluation has been overseen by a Reviewing Editor and a Senior Editor. The following individuals involved in review of your submission have agreed to reveal their identity: Guido Favia (Reviewer #1); Louis Lambrechts (Reviewer #3).

Our decision has been reached after consultation between the reviewers. Based on these discussions and the individual reviews below, we regret to inform you that your work will not be considered in its current form for publication in *eLife*.

The reviewers concur that the work presented has the potential to provide an interesting advancement of knowledge in the relationships between components of the midgut-associated fungi and mosquito's vector capacity.

However the study is preliminary in several respects and fails to provide evidence that gut digestive activity modulates dengue susceptibility. Additional experiments are required to establish a direct and functional link between DENV susceptibility and trypsin activity.

Two of the reviewers also raised serious concerns about the inappropriate statistical methods used to analysis of data and evaluate significance of the results. Moreover the raw data must be included to meet the standards in data reporting.

Despite being generally positive about the nature of this study the identified weaknesses will take some time to address (likely more than 2 months). While we are rejecting the paper as a result of this, I am broadly supportive of this manuscript and if you feel that you are able to address these issues we will consider a newly submitted form of this paper that will be treated as a revised manuscript.

Reviewer #1:

Rarely I have enjoyed so much to review a manuscript. This one deals with a very interesting subject and targeting the mycobiota component of mosquito vector is very original since most of the work performed up to now has been mainly targeted to bacteria and viruses.

The findings are relevant and supported by rigorous and coherent methodological approach. The Introduction describes the status of the field well, the results are overall well-presented, and the discussion is well-organized and thoughtful.

Materials and methods are well described allowing others to replicate the work.

Figures, tables and Supplementary Information are well designed and all reader to easily catch the essence of the results and their intrinsic originality.

The Bibliography is complete, providing references in an adequate number.

The authors addressed all the questions the work was raising (especially *Tsp_PR* secretome mediate the enhancement of DENV infection through a bacteria-independent mechanism). Ultimately, it is a really nice piece of science.

Reviewer #2:

The manuscript describes laboratory experiments that interrogate the impact of a *Talaromyces* fungus on *Aedes aegypti* susceptibility to dengue virus. Regretfully, in its current form, the manuscript is too preliminary for publication in *eLife*.

The title of the manuscript is mis-leading as there is no evidence that gut digestive activity modulates dengue susceptibility. The problem is that the authors used two unrelated systems to address their question; they first fed mosquitoes with fungal spores to examine the effect on dengue susceptibility in the midgut, fungal persistence throughout the duration of the experiment for mosquito survival assays. However, the rest of the manuscript is based on fungal secretome. This is like comparing apples and oranges. There is no evidence that spores develop further or persist as spores in the mosquito midgut. There is no evidence that the spores persist in the midgut as the experiments have been performed with whole mosquitoes.

Even should the assumption that spores develop as actively as on the fungal medium and secrete all factors, there is no direct evidence that trypsin inhibition underlies the observed susceptibility, there is no functional analyses of the causative effect of trypsin genes or trypsin activity.

Finally, as described below, the authors do not use appropriate statistical methods to evaluate the significance of their results, and they do not provide their raw data for confirmation.

Reviewer #3:

This experimental study provides evidence that a naturally occurring fungus isolated from the midgut of field-caught *Aedes aegypti* specimens can influence DENV infection and blood digestion in the midgut of mosquitoes previously exposed to fungal spores or fungus-secreted molecules. Overall, the paper is well structured and concisely written.

The main issue is the lack of direct evidence to link the anti-DENV effect and the digestion-disrupting effect of the fungus. Although the paper is entitled "An *Aedes aegypti*-associated fungus increases susceptibility to dengue virus by modulating gut digestive activity", the data only show that *Tsp_PR* (1) increases DENV titer in the *Aedes aegypti* midgut and (2) inhibits transcription and activity of blood-digesting enzymes. Whether (1) and (2) are related remains to be conclusively demonstrated. The causative link implied by the paper as it currently stands could be misleading.

[Editors’ note: what now follows is the decision letter after the authors submitted for further consideration.]

Thank you for submitting your article "An *Aedes aegypti*-associated fungus increases susceptibility to dengue virus by modulating gut trypsin activity" for consideration by *eLife*. Your article has been reviewed by four peer reviewers, and the evaluation has been overseen by a Reviewing Editor and Wendy Garrett as the Senior Editor. The following individuals involved in review of your submission have agreed to reveal their identity: Sassan Asgari (Reviewer #1); Fernando Noriega (Reviewer #3).

The reviewers have discussed the reviews with one another and the Reviewing Editor has drafted this decision to help you prepare a revised submission.

Overall the quality of the paper is compromised unless the conclusions are revised based on a more rigorous analysis and presentation of the data.

Summary:

In this study Anglero-Rodriguez et al. present evidence that a fungus originally isolated from wild *Aedes aegypti* mosquitoes can affect dengue virus infection and digestive processes when introduced into the midgut. This effect is credited to a down-regulation of genes involved in blood digestion, resulting in a decrease of trypsin activity, impaired digestion and diminished oogenesis. Apparently, trypsin also represses microbiota proliferation, as flies treated with the *Tsp_PR* secretome also harbor about 2x the density of bacteria in their gut.

The results, although mostly correlative, are significant. The molecular bases of the fungus effect are not explored in detail, but the study is s still comprehensive and might in part explain why certain populations of *Ae. aegypti* are more prolific vectors of dengue than are others.

Essential revisions:

One of the main criticisms of the previous version of the manuscript was that the study did not conclusively link the anti-dengue effect with the disruption of blood digestion. In the revised manuscript, the authors claim that this functional link is now demonstrated by additional data shown in Figure 7.

Throughout the paper there are statistical issues with the use of ANOVA to analyze virus titers (log PFU/midgut). These analyses include virus-negative mosquitoes and this has two undesirable consequences. First, the residuals of the analysis are typically not normally distributed, which violates one of the fundamental assumptions of ANOVA. Second, the analysis of infection intensity is partially redundant with that of prevalence because it includes the effect of infection status (infected or uninfected).

Figure 2, Figure 3, Figure 5, Figure 7. The upper right infection prevalence boxes were not statistically analyzed (or the stats aren't shown) to indicate if a significant difference was observed. Thus, the authors can't technically state that they observed an increase in infection prevalence (e.g., for Figure 2, second paragraph of the Results section).

Figure 7. Ovary development is a tangential parameter for quantifying blood meal digestion. Other factors could contribute to the reduced ovary size observed in *Tsp_PR* secretome treated mosquitoes. For example, *Tsp_PR* secretome treated mosquitoes house significantly more bacteria in their gut, and this population may include taxa not normally present. This could induce an immune response. The reduced ovary development observed under these circumstances could reflect a competition for resources between the bacteria and the mosquito. A more direct measurement of blood meal digestion is to simply weight the gut (as in Emre Aksoy et al. 2016, PNAS) or better yet, take photos (as in Bryant et al., 2010).

Figure 7 the authors use ANOVA to compare log-transformed virus titers per mosquito, across several trypsin knockdown treatments. They include virus-negative mosquitoes in their analysis, which results in the shortcomings mentioned above. When one performs the ANOVA of virus titers without virus-negative mosquitoes, for T714 and Tmix (the treatments with the strongest effect according to the authors) the residuals meet the normality assumption of ANOVA and there is a statistically significant interaction between experiment and treatment. In fact, the treatment effect is only seen for the first experiment, but not for the other two experiments. This means that the effect is inconsistent and seriously questions the validity of the conclusion.

Figure 7 the authors analyze infection prevalence across several trypsin knockdown treatments. In the Results section the authors claim "silencing of T714 resulted in the greatest increase of DENV infection prevalence". This sentence together with the blow-up view of infection percentages (without plotted confidence intervals) in Figure 7 are strongly misleading. Even for T714 there is no statistically significant effect of the treatment on infection prevalence.

---

## [Author Response]

[Editors’ note: the author responses to the first round of peer review follow.]

Reviewer #2:The manuscript describes laboratory experiments that interrogate the impact of a Talaromyces fungus on Aedes aegypti susceptibility to dengue virus. Regretfully, in its current form, the manuscript is too preliminary for publication in eLife.The title of the manuscript is mis-leading as there is no evidence that gut digestive activity modulates dengue susceptibility. The problem is that the authors used two unrelated systems to address their question; they first fed mosquitoes with fungal spores to examine the effect on dengue susceptibility in the midgut, fungal persistence throughout the duration of the experiment for mosquito survival assays. However, the rest of the manuscript is based on fungal secretome. This is like comparing apples and oranges. There is no evidence that spores develop further or persist as spores in the mosquito midgut. There is no evidence that the spores persist in the midgut as the experiments have been performed with whole mosquitoes.

We don’t agree with the apples and oranges metaphor used by the reviewer; apples and apple juice would have been more accurate. The first observation of the study was that feeding on live fungi resulted in an enhanced DENV infection, next we show that the effect on infection is attributed to fungus secreted factors. Fungi are generally known for their production and secretion of a variety of bioactive secondary metabolites. We proceeded with the remaining experiments using the fungus secretome because it contains the bioactivity of interest and represented a less complex sample than the entire organism, and the effect was stronger (*P* <0.0001). The probability that the effect on DENV infection exerted by the spores/conidia (which we have shown secreted the bioactive molecule(s) which we call “secretome”) when present the midgut is attributed to something different from that contained in the filtered spore/conidia culture (secretome) is so small, and difficult to address, that we don’t think it deserved further speculation. We have addressed the other concerns regarding spores persistence in the midgut by performing a new set of experiments where fungus persistence up to 25 days after ingestion was monitored. We now show the persistence of the conidia in the mosquito midgut, after a single feeding on conidia, throughout the entire time-course.

Regarding the title of the manuscript; we have now provided functional evidence, through genesilencing assays, linking decreased trypsin transcript abundance to enhanced DENV infection (Figure 7).

Even should the assumption that spores develop as actively as on the fungal medium and secrete all factors, there is no direct evidence that trypsin inhibition underlies the observed susceptibility, there is no functional analyses of the causative effect of trypsin genes or trypsin activity.

In the original manuscript we presented functional assays showing that midguts treated with the fungus secretome display a significantly lower trypsin enzymatic activity (now Figure 7).

However, to further address the reviewer’s concern, we have now provided functional evidence, through gene-silencing assays, linking decreased trypsin transcript abundance to enhanced DENV infection (Figure 7).

Finally, as described below, the authors do not use appropriate statistical methods to evaluate the significance of their results, and they do not provide their raw data for confirmation.

We have provided a detailed raw data file outlining the statistical analyses.

[Editors' note: the author responses to the re-review follow.]

Essential revisions:One of the main criticisms of the previous version of the manuscript was that the study did not conclusively link the anti-dengue effect with the disruption of blood digestion. In the revised manuscript, the authors claim that this functional link is now demonstrated by additional data shown in Figure 7.

In the first revision, we did not claim that the anti-dengue effect is linked with the disruption of blood digestion, but with a disruption of trypsin activity. We explained the reduced ovary development as a possible result of impaired blood digestion. We clarify this aspect in the revised manuscript, and included additional data as suggested by the reviewer, showing that mosquitoes treated with fungus secretome weight more than untreated mosquitoes at 48h post-blood meal (now Figure 7).

Throughout the paper there are statistical issues with the use of ANOVA to analyze virus titers (log PFU/midgut). These analyses include virus-negative mosquitoes and this has two undesirable consequences. First, the residuals of the analysis are typically not normally distributed, which violates one of the fundamental assumptions of ANOVA. Second, the analysis of infection intensity is partially redundant with that of prevalence because it includes the effect of infection status (infected or uninfected).

We eliminated virus-negative mosquitoes from the intensity graphs and only include this information in graphs showing infection prevalence. We sought specialist advice on statistical analyses from the Johns Hopkins School of Public Health Biostatistics department, and they analyzed the data using Generalized Linear Regression (GLM) with experiment-clustered robust variance estimates to account for potential within-experiment correlation of outcomes (Rogers, 1993) see Statistical Analysis in the Materials and methods section for more details.

Figure 2, Figure 3, Figure 5, Figure 7. The upper right infection prevalence boxes were not statistically analyzed (or the stats aren't shown) to indicate if a significant difference was observed. Thus, the authors can't technically state that they observed an increase in infection prevalence (e.g., for Figure 2, second paragraph of the Results section).

We performed statistical analysis of the prevalence data, and include this information in the revised manuscript.

Figure 7. Ovary development is a tangential parameter for quantifying blood meal digestion. Other factors could contribute to the reduced ovary size observed in Tsp_PR secretome treated mosquitoes. For example, Tsp_PR secretome treated mosquitoes house significantly more bacteria in their gut, and this population may include taxa not normally present. This could induce an immune response. The reduced ovary development observed under these circumstances could reflect a competition for resources between the bacteria and the mosquito. A more direct measurement of blood meal digestion is to simply weight the gut (as in Emre Aksoy et al. 2016, PNAS) or better yet, take photos (as in Bryant et al., 2010).

Ovary development is part of the gonotrophic cycle after a mosquito takes a blood meal, and impairment of ovary development can indicate compromised blood digestion or nutrient acquisition (Bryant et al., 2010; Lea et al., 1978). However, as suggested by the reviewer, and used by others (Pimenta de Oliveira et al., 2017), we now present new data on mosquito body weight after blood feeding, showing that mosquitoes treated with fungus secretome are heavier than the non-treated at 48h post-blood meal suggesting a compromised blood degradation process (Figure 7).

Figure 7 the authors use ANOVA to compare log-transformed virus titers per mosquito, across several trypsin knockdown treatments. They include virus-negative mosquitoes in their analysis, which results in the shortcomings mentioned above. When one performs the ANOVA of virus titers without virus-negative mosquitoes, for T714 and Tmix (the treatments with the strongest effect according to the authors) the residuals meet the normality assumption of ANOVA and there is a statistically significant interaction between experiment and treatment. In fact, the treatment effect is only seen for the first experiment, but not for the other two experiments. This means that the effect is inconsistent and seriously questions the validity of the conclusion.

We eliminated the virus-negative mosquitoes for all graphs and only included this information in the prevalence graphs. We also included a 4^th^ independent experiment to the data set to render it more robust. Data was analyzed using Generalized Linear Regression (GLM) with experiment-clustered robust variance estimates to account for potential within-experiment correlation of outcomes (Rogers, 1993) see Statistical Analysis in the Materials and methods section for more details.

Figure 7 the authors analyze infection prevalence across several trypsin knockdown treatments. In the Results section the authors claim "silencing of T714 resulted in the greatest increase of DENV infection prevalence". This sentence together with the blow-up view of infection percentages (without plotted confidence intervals) in Figure 7 are strongly misleading. Even for T714 there is no statistically significant effect of the treatment on infection prevalence.

For clarity, we decided to show each trypsin silencing graph in an individual panel with its respective prevalence value and confidence intervals error bars.